# DECOUPLED DIFFUSION MODELS: IMAGE TO ZERO AND ZERO TO NOISE

## ABSTRACT

This paper proposes decoupled diffusion models (DDMs), featuring a new diffusion paradigm that allows for high-quality (un)conditioned image generation in less than 10 function evaluations. In a nutshell, DDMs decouple the forward image-to-noise mapping into *image-to-zero* mapping and *zero-to-noise* mapping. Under this framework, we mathematically derive 1) the training objectives and 2) for reverse time the sampling formula based on an analytic transition probability which models image to zero transition. The former enables DDMs to learn noise and image components separately which simplifies learning. Importantly, because of the latter's analyticity in the *zero-to-image* sampling function, DDMs can avoid the ordinary differential equation based accelerators and instead naturally perform sampling with an arbitrary step size. Under the few function evaluation setup, DDMs experimentally yield very competitive performance compared with the state of the art in 1) unconditioned image generation, *e.g.*, CIFAR-10 and CelebA-HQ-256 and 2) image-conditioned downstream tasks such as super-resolution, saliency detection, and image inpainting.

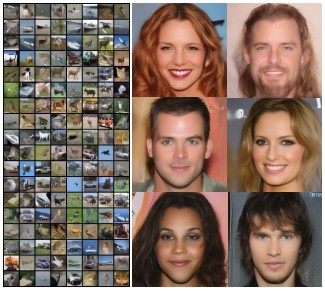
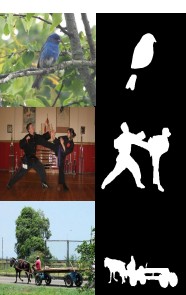
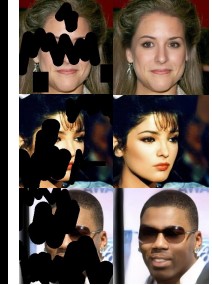
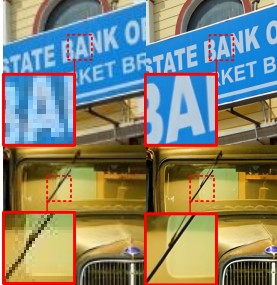

(a) Unconditioned generation (10 steps)   (b) Conditioned generation (5 steps)

Figure 1: High-quality images generated by the proposed DDMs under few-step settings. (a) 10-step unconditioned generation on the CIFAR-10 and CelebA-HQ-256 datasets. (b) 5-step conditioned tasks including saliency detection, image inpainting and super-resolution.

## 1 INTRODUCTION

Diffusion probabilistic models (DPMs) (Ho et al., 2020; Nichol & Dhariwal, 2021; Ho & Salimans, 2022; Dhariwal & Nichol, 2021) have made impressive achievements in content generation (Karras et al., 2022; Chen et al., 2021a;b). DPMs view the image-to-noise process as a parameterized Markov chain that gradually adds noise to the original data until the signal is completely corrupted and uses a neural network to model the reversed denoising process for new sample generation.

Song et al. (2021c) prove that the forward diffusion process of DPMs is equivalent to the score-based generative models (SGMs) (Sohl-Dickstein et al., 2015; Song et al., 2021b; Kingma et al., 2021). Both can be represented by a stochastic differential equation (SDE). Essentially, training of DPMs is to learn the score function, gradient of the log probability density, of the perturbed data. Song

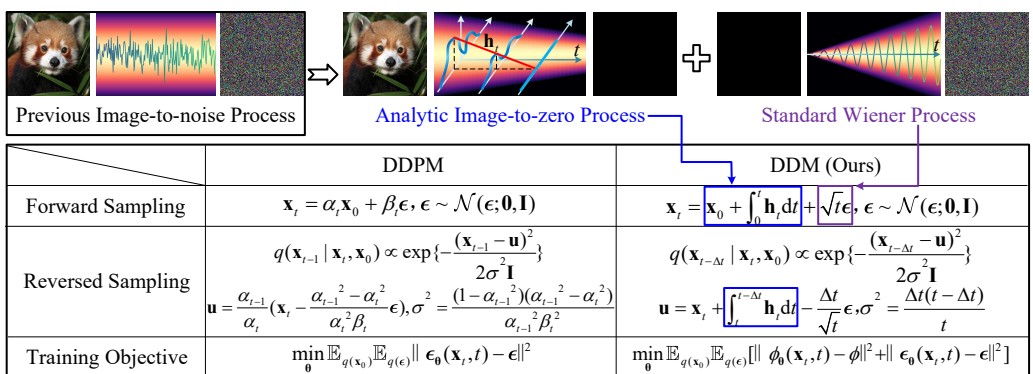

Figure 2: Framework overview. (**Top**:) DPMs typically use an image-to-noise process, while we propose to split it into two relatively simpler processes: the image-to-zero mapping and the zero-to-noise mapping. We use an *analytic* function (in blue boxes) to model image to zero, or the attenuation gradient of the image (red line in the middle image). The zero-to-noise path is governed by the standard Wiener process. (**Bottom**:) We compare the equations of forward sampling, reversed sampling, and training objective of our method and DDPM.

et al. (2021c) show that the learned score function is uniquely dependent on the forward diffusion process. Therefore, the forward diffusion process is critical for the effectiveness of DPMs.

This paper proposes decoupled diffusion models (DDMs). Instead of using the normal image-to-noise forward diffusion, we decouple this process: attenuate the image component by an image-to-zero mapping and increase the noise component by a zero-to-noise mapping. To model the latter, we use the standard Wiener process (Einstein et al., 1905). To model the former, we design various analytic functions such as $\mathbf{h}_t = \mathbf{c}$. In this framework, we derive our training objective: instead of predicting only noise from $\mathbf{x}_t$, we predict the clean image and noise independently from $\mathbf{x}_t$. We derive the sampling formula based on the analytic transition function for reversed diffusion.

The designed framework has two benefits. First, by splitting image-to-noise mapping into two relatively simpler processes, we empirically find it improves the effectiveness of diffusion model training. In Fig. 4, either adding a branch to predict the image component or decoupling forward diffusion into two branches allows the denoising process to achieve high quality with fewer evaluations. Second, during reversed diffusion, the image component can be obtained by an *analytical* solution, which allows the reversed-time SDE to be solved with an arbitrary step size, thus significantly reducing the number of function evaluations. This benefit is ensured by the use of an *analytic* transition function for modeling image-to-zero mapping. This offers a significant advantage over typical DPMs, where solving the reverse-time process accurately requires numerical integration, resulting in thousands of function evaluations. We summarize the main points of this paper below.

- We introduce decoupled diffusion models (DDMs), where the image-to-noise process is decoupled into image-to-zero and zero-to-noise processes.
- We derive the training objective and analytical forward/reversed SDE. Via the former, DDMs learn to predict the image and noise simultaneously from $\mathbf{x}_t$. Via the latter, we avoid time-consuming numerical integration and achieve high-quality image generation with much fewer steps.
- DDMs yield competitive performance in unconditional image generation under few function evaluation setup. Moreover, DDMs can be easily applied to downstream image-conditioned applications such as image super-resolution, depth estimation, saliency detection, and image inpainting, and give very competitive accuracy within 10 function evaluations.

## 2 RELATED WORK

**Diffusion probabilistic models** have demonstrated outstanding capabilities in generating images (Dhariwal & Nichol, 2021; Ho et al., 2022), speech and music (Mittal et al., 2021), and 3D shapes

(Müller et al., 2023; Wang et al., 2023). Sohl-Dickstein et al. (2015) formulated diffusion probabilistic models, and Ho et al. (2020) first proposed a general diffusion framework for computer vision applications. Song et al. (2021c) established a link between DPMs and score-based generative models and derived general SDE frameworks for both the forward and reversed processes. Later, Rombach et al. (2022) and Vahdat et al. (2021) translated the diffusion process from the image space to latent space with an auto-encoder, allowing the model to generate higher-resolution images. Their attempts mainly translate the data space into a smoother one instead of improving the diffusion process itself. For the latter, Dockhorn et al. (2022a) introduce velocity variables for the diffusion process for better performance and saving sampling time. They couple velocity with the image space, resulting in a more complex diffusion process. In comparison, we propose decoupling the diffusion process into two relatively simpler processes, where an analytic function is used for modeling the image-to-zero process, allowing for faster reversed diffusion.

**Few-step generation with DPMs** aims to accelerate the sampling process. One line of works (Song et al., 2021a; Lu et al., 2022; Faradonbeh et al., 2022; Doucet et al., 2022; Dockhorn et al., 2022b) construct ODE solvers of one or higher order to accelerate numerical integration in the reversed process. Essentially, they do not change the diffusion paradigm. A potential downside is that samples generated from these ODE solvers are determined by the initial noise thus exhibiting less diversity. In the other line of works, distillation-based methods (Zhang & Chen, 2022; Salimans & Ho, 2022; Song et al., 2023; Meng et al., 2022; Luhman & Luhman, 2021) use a fitted teacher model to distill the student model, allowing the latter to generate with few steps. In these methods, training is usually divided into a few stages, which increases the training cost. Differently, this paper does not focus on improving the sampling algorithms or training skills: the proposed DDM is an enhanced diffusion method that naturally supports generating high-quality images with fewer steps.

## 3 PRELIMINARIES

The forward diffusion process of a typical DPM (Sohl-Dickstein et al., 2015) is described as a Markov chain. Consider a continuous-time Markov chain with $t \in [0, 1]$:

$$q(\mathbf{x}_t|\mathbf{x}_0) = \mathcal{N}(\mathbf{x}_t; \alpha_t \mathbf{x}_0, \beta_t^2 \mathbf{I}), \tag{1}$$

where $\alpha_t, \beta_t$ are differentiable functions of time $t$ with bounded derivatives. $\beta_t$ is designed to increase gradually over time while $\alpha_t$ does the opposite, ensuring $q(\mathbf{x}_1|\mathbf{x}_0) = \mathcal{N}(\mathbf{x}_1; \mathbf{0}, \mathbf{I})$. It is proven that this Markov chain can be represented by the following SDE (Kingma et al., 2021):

$$d\mathbf{x}_t = f_t \mathbf{x}_t dt + g_t d\mathbf{w}_t, \quad \mathbf{x}_0 \sim q(\mathbf{x}_0), \tag{2}$$

where $f_t = \frac{d \log \alpha_t}{dt}, g_t^2 = \frac{d\beta_t^2}{dt} - 2f_t \beta_t^2$, and $\mathbf{w}_t$ is the standard Wiener process. Song et al. (2021c) shows that the corresponding reversed SDE inverting $\mathbf{x}_1$ to $\mathbf{x}_0$ can be derived as:

$$d\mathbf{x}_t = [f_t \mathbf{x}_t - g_t^2 \nabla_{\mathbf{x}} \log q(\mathbf{x}_t)]dt + g_t d\overline{\mathbf{w}}_t, \tag{3}$$

where $\overline{\mathbf{w}}_t$ is a standard Wiener process in the reversed diffusion. The only unknown term in Eq. 3, $\nabla_{\mathbf{x}} \log q(\mathbf{x}_t)$, is given by a neural network $\epsilon_{\boldsymbol{\theta}}(\mathbf{x}_t, t)$ parameterized by $\boldsymbol{\theta}$. $\epsilon_{\boldsymbol{\theta}}(\mathbf{x}_t, t)$ estimates the scaled score function, i.e., $-\beta_t \nabla_{\mathbf{x}} \log q(\mathbf{x}_t)$, and the training objective (Ho et al., 2020) is:

$$
\begin{aligned}
\mathcal{L}(\boldsymbol{\theta}, \lambda_t) &:= \int_0^1 \lambda_t \mathbb{E}_{q(\mathbf{x}_0)} \mathbb{E}_{q(\boldsymbol{\epsilon})}[\|\epsilon_{\boldsymbol{\theta}}(\mathbf{x}_t, t) + \beta_t \nabla_{\mathbf{x}} \log q(\mathbf{x}_t)\|^2]dt \\
&= \int_0^1 \lambda_t \mathbb{E}_{q(\mathbf{x}_0)} \mathbb{E}_{q(\boldsymbol{\epsilon})}[\|\epsilon_{\boldsymbol{\theta}}(\mathbf{x}_t, t) - \boldsymbol{\epsilon}\|^2]dt,
\end{aligned}
\tag{4}
$$

where $\boldsymbol{\epsilon} \sim \mathcal{N}(\boldsymbol{\epsilon}; \mathbf{0}, \mathbf{I}), \mathbf{x}_t = \alpha_t \mathbf{x}_0 + \beta_t \boldsymbol{\epsilon}$, and $\lambda_t$ is a weighting function.

Replacing the score function by $\epsilon_{\boldsymbol{\theta}}(\mathbf{x}_t, t)$ in Eq. 3, one can generate samples by solving the following reversed SDE with numerical integration, starting from $\mathbf{x}_1 \sim \mathcal{N}(\mathbf{x}_1; \mathbf{0}, \mathbf{I})$:

$$d\mathbf{x}_t = [f_t \mathbf{x}_t + \frac{g_t^2}{\beta_t} \epsilon_{\boldsymbol{\theta}}(\mathbf{x}_t, t)]dt + g_t d\overline{\mathbf{w}}_t. \tag{5}$$

Due to the complexity of the SDE, it is difficult to estimate the score function accurately and let alone finding an analytical solution for the reverse-time SDE. As a result, DPMs need thousands of function evaluations to generate high-quality images, resulting in extremely long inference time.

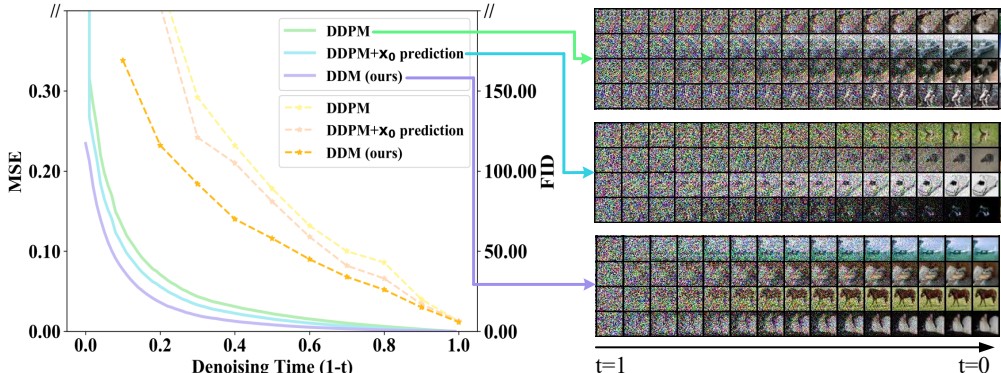

Figure 3: Comparing DDPM, its improved version, and DDM of image quality of unconditioned generation on CIFAR-10. (**Left**:) We evaluate image quality at each step using mean square error (MSE) of the final $\mathbf{x}_0$ ($t = 0$) and the estimated $\hat{\mathbf{x}}_0$ during denoising time. (**Right**:) Sample generated images from $t = 0.4$ for the three compared methods. Improvement can be observed after adding to DDPM a branch that predicts $\mathbf{x}_0$. Further improvement is confirmed when the image-to-noise process is completely decoupled to predict $\mathbf{x}_0$ and noise simultaneously.

## 4 METHOD

### 4.1 PRELIMINARY EXPERIMENTS: IMPROVING DDPM WITH $\mathbf{x}_0$ PREDICTION

To show the benefit of decoupling the forward diffusion, we use DDPM (Ho et al., 2020) as baseline and create an improved version. The two methods are described below.

- DDPM. It has a small U-net architecture (48M parameters) and can be trained using only the noise loss (Ho et al., 2020). Because this paper uses continuous time $t \in [0, 1]$, we linearly transform the discrete time $t = 1, 2, ..., 1000$ used in DDPM into $t \in [0, 1]$ with interval 0.001.

- DDPM+$\mathbf{x}_0$ prediction. On top of DDPM, we add a convolutional branch to predict $\mathbf{x}_0$. Details of this branch are provided in the supplementary materials.

In Fig. 3, we compare the unconditioned image generation quality of the three methods. Specifically, after obtaining an estimation of $\mathbf{x}_0$ at any time $t$ in the reverse-time process, we use the mean squared error (MSE) between the estimated $\hat{\mathbf{x}}_0$ and $\mathbf{x}_0$ to indicate image quality generated at time $t$.

From Fig. 3, at $t \approx 0.5$, images generated by the vanilla DDPM are quite close to noise. In fact, Fig. 3 shows that it usually takes $900 \sim 1000$ steps ($t = 0.1 \sim 0$) for DDPM to generate meaningful images. In comparison, quality of images generated by "DDPM+$\mathbf{x}_0$ prediction" is much higher than the vanilla DDPM. It suggests that predicting both noise and $\mathbf{x}_0$ from $\mathbf{x}_t$ with additional $\mathbf{x}_0$ supervision benefits diffusion model training.

While these experiments indicate the usefulness of $\mathbf{x}_0$ prediction, in DDPM, $\mathbf{x}_0$ is still deeply coupled with noise in $\mathbf{x}_t$. This consideration motivates us to explicitly and separately model $\mathbf{x}_0$ along time. In fact, using the proposed $\mathbf{x}_0$ modeling method DDM[1] with the same U-Net structure and noise loss as "DDPM+$\mathbf{x}_0$ prediction", image generation quality is further improved (see Fig. 3).

### 4.2 PROPOSED DECOUPLED DIFFUSION MODELS

This section introduces a forward process and a reversed process, representing the pathways of image corruption and denoising, respectively. We prove that the proposed decoupled forward process is equivalent to the previous mapping from image to noise. We then derive the training objective of DDM. Finally, we use the Bayesian theorem to derive the sampling formula of DDM which, due to its analytic nature, allows for sampling with arbitrary step sizes and thus few-step generation.

---

[1]Details of DDM used in this preliminary experiment are provided in Appendix B

**Forward process**. We propose a new diffusion formula to describe the forward process:

$$\mathbf{x}_t = \mathbf{x}_0 + \int_0^t \mathbf{h}_t \mathrm{d}t + \int_0^t \mathrm{d}\mathbf{w}_t, \quad \mathbf{x}_0 \sim q(\mathbf{x}_0). \tag{6}$$

Here, $\mathbf{x}_0 + \int_0^t \mathbf{h}_t \mathrm{d}t$ describes the image attenuation process, *i.e.*, image to zero, and $\int_0^t \mathrm{d}\mathbf{w}_t$ denotes the noise increasing process, *i.e.*, zero to noise. $\mathbf{h}_t$ is a differentiable function of $t$, and $\mathbf{w}_t$ is the standard Wiener process. Eq. 6 shows that $q(\mathbf{x}_t|\mathbf{x}_0)$ can be regarded as a normal distribution with mean $\mathbf{x}_0 + \int_0^t \mathbf{h}_t \mathrm{d}t$ and variance $t\mathbf{I}$. We prove that the proposed decoupled diffusion process defined in Eq. 6 is equivalent to previous SDE in Appendix A.

The forward process must ensure that $\mathbf{x}_t \sim q(\mathbf{x}_0)$ when $t = 0$ and $\mathbf{x}_t \sim \mathcal{N}(\mathbf{x}_1; \mathbf{0}, \mathbf{I})$ when $t = 1$. Obviously, Eq. 6 satisfies the first requirement when setting $t = 0$. Moreover, the noise term in Eq. 6 itself meets the second requirement because $\int_0^t \mathrm{d}\mathbf{w}_t = \mathbf{w}_t - \mathbf{w}_0 \sim \mathcal{N}(\mathbf{0}, t\mathbf{I})$. So, we only need to design $\mathbf{h}_t$ such that: $\mathbf{x}_0 + \int_0^1 \mathbf{h}_t \mathrm{d}t = \mathbf{0}$. Note that $\mathbf{h}_t$ *can be analytic, which will significantly improve the efficiency of the reversed process*. We analyze various forms of $\mathbf{h}_t$ in Section 5.

For simplicity, we denote $\mathbf{H}_t = \int_0^t \mathbf{h}_t \mathrm{d}t$ and give the distribution of $\mathbf{x}_t$ conditioned on $\mathbf{x}_0$:

$$q(\mathbf{x}_t|\mathbf{x}_0) = \mathcal{N}(\mathbf{x}_t; \mathbf{x}_0 + \mathbf{H}_t, t\mathbf{I}). \tag{7}$$

Thus, we can sample $\mathbf{x}_t$ by $\mathbf{x}_t = \mathbf{x}_0 + \mathbf{H}_t + \sqrt{t}\boldsymbol{\epsilon}$ in the forward process, where $\boldsymbol{\epsilon} \sim \mathcal{N}(\mathbf{0}, \mathbf{I})$.

**Reversed process**. Based on the the distribution of $\mathbf{x}_t$ conditioned on $\mathbf{x}_0$, we build the reversed diffusion process using the Bayesian theorem. Unlike DDPM (Ho et al., 2020) which uses *discrete*-time Markov chain, we employ *continuous*-time Markov chain with the smallest time step $\Delta t \to 0^+$ and use conditional distribution $q(\mathbf{x}_{t-\Delta t}|\mathbf{x}_t, \mathbf{x}_0)$ to approximate $q(\mathbf{x}_{t-\Delta t}|\mathbf{x}_t)$:

$$q(\mathbf{x}_{t-\Delta t}|\mathbf{x}_t, \mathbf{x}_0) = \frac{q(\mathbf{x}_t|\mathbf{x}_{t-\Delta t}, \mathbf{x}_0)q(\mathbf{x}_{t-\Delta t}|\mathbf{x}_0)}{q(\mathbf{x}_t|\mathbf{x}_0)}. \tag{8}$$

Given the forward diffusion formula, we derive the reversed transition probability $q(\mathbf{x}_{t-\Delta t}|\mathbf{x}_t, \mathbf{x}_0)$ also follows a normal distribution:

$$
\begin{aligned}
q(\mathbf{x}_{t-\Delta t}|\mathbf{x}_t, \mathbf{x}_0) &\propto \exp\{-\frac{(\mathbf{x}_{t-\Delta t} - \widetilde{\mathbf{u}})^2}{2\widetilde{\sigma}^2 \mathbf{I}}\}, \\
\widetilde{\mathbf{u}} &= \mathbf{x}_t + \mathbf{H}_{t-\Delta t} - \mathbf{H}_t - \frac{\Delta t}{\sqrt{t}}\boldsymbol{\epsilon}, \\
\widetilde{\sigma}^2 &= \frac{\Delta t(t - \Delta t)}{t},
\end{aligned}
\tag{9}
$$

where $\boldsymbol{\epsilon} \sim \mathcal{N}(\mathbf{0}, \mathbf{I})$. In Eq. 9, the variance term $\widetilde{\sigma}^2$ is non-parametric and solely dependent on the current time $t$ and the step size $\Delta t$, whereas the mean term $\widetilde{\mathbf{u}}$ involves two unknown variables: $\mathbf{H}_t$ and $\boldsymbol{\epsilon}$ in the reversed process. As a result, to solve the reversed process, we must use $p_{\boldsymbol{\theta}}(\mathbf{x}_{t-\Delta t}|\mathbf{x}_t)$ to approximate $q(\mathbf{x}_{t-\Delta t}|\mathbf{x}_t, \mathbf{x}_0)$ and estimate $\mathbf{H}_t$ and $\boldsymbol{\epsilon}$ using a neural network. Essentially, the value of $\mathbf{H}_t$ depends on $\mathbf{h}_t$, an analytic function with respect to $t$, which is determined by its parameters $\phi$. The ground truth of $\phi$ can be obtained by solving $\mathbf{x}_0 + \int_0^1 \mathbf{h}_t \mathrm{d}t = \mathbf{0}$ and details can be found in Appendix B. In training, we predict $\phi$ and $\boldsymbol{\epsilon}$ by the U-net architecture with two decoder branches: $\phi_{\boldsymbol{\theta}}, \boldsymbol{\epsilon}_{\boldsymbol{\theta}} = \mathrm{Net}_{\boldsymbol{\theta}}(\mathbf{x}_t, t)$. We refer readers to Appendix A and B for more details about derivations and U-Net architectures.

**Training objective.** By maximizing the evidence lower bound of the log-likelihood, we demonstrate the training objective of DDM (please refer to Appendix A for details):

$$\min_{\boldsymbol{\theta}} \mathbb{E}_{q(\mathbf{x}_0)}\mathbb{E}_{q(\boldsymbol{\epsilon})}[\|\phi_{\boldsymbol{\theta}} - \phi\|^2 + \|\boldsymbol{\epsilon}_{\boldsymbol{\theta}} - \boldsymbol{\epsilon}\|^2]. \tag{10}$$

In Eq. 10, the first term and second term correspond to the image component and noise component, respectively. That is, the model is trained to predict image and noise separately from $\mathbf{x}_t$, in line with our objective of decoupling the clean image and noise components from $\mathbf{x}_t$. We attach two hyperparameters $\lambda_1$ and $\lambda_2$ to the first and second terms of Eq. 10, respectively. During the reverse-time process, the noise represents the starting point for $\mathbf{x}_t$, whereas the clean image represents the endpoint. Simultaneously predicting the image and noise components allows $\mathbf{x}_t$ to be aligned with both starting and ending points at each time step. Detailed training procedure is presented in Alg. 1.

| **Algorithm 1** Training algorithm of DDM. | **Algorithm 2** Sampling algorithm of DDM. |
|---|---|
| 1: **Initialize** $i = 0, N = num\_iters, lr,$ network parameters: $\boldsymbol{\theta}$ | 1: **Initialize** $\mathbf{x}_t \sim \mathcal{N}(\mathbf{0}, \mathbf{I}), t = 1, \quad N = num\_steps, s = 1/N, \text{Net}_{\boldsymbol{\theta}};$ |
| 2: **while** $i < N$ **do** | 2: **while** $t > 0$ **do** |
| 3: $\quad t \sim Uniform(0,1), \mathbf{x}_0 \sim q(\mathbf{x}_0), \boldsymbol{\epsilon} \sim \mathcal{N}(\mathbf{0}, \mathbf{I}), \phi \sim \mathbf{x}_0 + \mathbf{H}_1 = \mathbf{0};$ | 3: $\quad \phi_{\boldsymbol{\theta}}, \boldsymbol{\epsilon}_{\boldsymbol{\theta}} = \text{Net}_{\boldsymbol{\theta}}(\mathbf{x}_t, t);$ |
| 4: $\quad \mathbf{H}_t = \int_0^t \mathbf{h}_t^{\phi} \mathrm{d}t;$ | 4: $\quad \mathbf{H}_t = \int_0^t \mathbf{h}_t^{\phi} \mathrm{d}t;$ |
| 5: $\quad \mathbf{x}_t = \mathbf{x}_0 + \mathbf{H}_t + \sqrt{t}\boldsymbol{\epsilon};$ | 5: $\quad \widetilde{\mathbf{u}} = \mathbf{x}_t + \mathbf{H}_{t-s} - \mathbf{H}_t - \frac{s}{\sqrt{t}}\boldsymbol{\epsilon}_{\boldsymbol{\theta}};$ |
| 6: $\quad \phi_{\boldsymbol{\theta}}, \boldsymbol{\epsilon}_{\boldsymbol{\theta}} = \text{Net}_{\boldsymbol{\theta}}(\mathbf{x}_t, t);$ | 6: $\quad \widetilde{\sigma} = \sqrt{\frac{s(t-s)}{t}};$ |
| 7: $\quad \boldsymbol{\theta} \leftarrow lr * \nabla_{\boldsymbol{\theta}}(\|\phi_{\boldsymbol{\theta}} - \phi\|^2 + \|\boldsymbol{\epsilon}_{\boldsymbol{\theta}} - \boldsymbol{\epsilon}\|^2);$ | 7: $\quad \mathbf{x}_t = \widetilde{\mathbf{u}} + \widetilde{\sigma} * \widetilde{\boldsymbol{\epsilon}}, \ \widetilde{\boldsymbol{\epsilon}} \sim \mathcal{N}(\mathbf{0}, \mathbf{I});$ |
| 8: $\quad i = i + 1;$ | 8: $\quad t = t - s;$ |
| 9: **end while**; | 9: **end while**; |
| 10: **return** $\boldsymbol{\theta};$ | 10: **return** $\mathbf{x}_t;$ |

**Few-step sampling from DDM.** After introducing the reverse-time process in the Section 4.2, we now directly sample from DDM by iteratively solving Eq. 9 from $t = 1$ to 0. Because $\mathbf{h}_t$ has an analytic form, DDM possesses the natural property of sampling with an arbitrary step size $s$. For example, under linear image-to-zero assumption $\mathbf{h}_t = \mathbf{c}$, $\mathbf{H}_t = \int_0^t \mathbf{h}_t \mathrm{d}t = \mathbf{c}t$; as such, we can easily obtain $\mathbf{H}_{t-s} = \mathbf{c}(t - s)$. Consequently, we can derive the expression for $q(\mathbf{x}_{t-s}|\mathbf{x}_t, \mathbf{x}_0)$ as:

$$
\begin{aligned}
q(\mathbf{x}_{t-s}|\mathbf{x}_t, \mathbf{x}_0) &\propto \exp\{-\frac{(\mathbf{x}_{t-s} - \widetilde{\mathbf{u}})^2}{2\widetilde{\sigma}^2 \mathbf{I}}\}, \\
\widetilde{\mathbf{u}} &= \mathbf{x}_t + \mathbf{H}_{t-s} - \mathbf{H}_t - \frac{s}{\sqrt{t}}\boldsymbol{\epsilon}, \\
\widetilde{\sigma}^2 &= \frac{s(t - s)}{t}.
\end{aligned}
\tag{11}
$$

Compared with DPMs where the sampling step size has to be $s = 1$, Eq. 11 allows $s$ to take an arbitrary value in $[0, 1]$. This is attributed to the analytic form of $\mathbf{h}_t$, which describes the proposed image-to-zero process. Technically speaking, Eq. 11 even allows us to generate images in a single step by directly setting $s = t = 1$. Nevertheless, it is challenging to give a good estimate of $\phi$ at the starting time. Moreover, one-step generation ($s = t$) causes the variance $\widetilde{\sigma}^2$ in Eq. 11 to be zero. The two problems make the generated images appear blurry and lack diversity. Therefore, we still sample iteratively but use a much larger step size than DPMs: this drastically reduces the sampling steps from 1000 to 10 while maintaining relatively good image quality. Specifically, we commence at $t = 1$ and take uniform steps of size $s$ until we reach $t = 0$. The algorithmic details are outlined in Alg. 2. Detailed proof of Eq. 11 is provided in Appendix A.

## 4.3 DISCUSSIONS

**Why decoupling the image-to-noise process into image to zero and zero to noise eases training?** In the image-to-noise process, estimating the noise component from $\mathbf{x}_t$, which is deeply mixed by $\mathbf{x}_0$ and noise, is not an easy task. Compared to image-to-noise process that only uses the noise loss to supervise network, image-to-zero and zero-to-noise mappings add the $\mathbf{x}_0$ modeling branch and use the noise and $\mathbf{x}_0$ to supervise the network simultaneously, thus reducing learning difficulty.

**Why this decoupling allows for fewer function evaluations?** Previous DPMs directly model the image-to-noise process, which requires numerical integration with very small step size since it is not analytic. Differently, the DDM has an analytic image-to-zero process, allowing the reversed process to be solved with arbitrary step sizes, so we can generate samples with fewer function evaluations.

**Establishing a new diffusion framework vs. fast sampling.** Our method is not directly comparable but complementary with fast sampling methods. DDMs benefit from the proposed analytic image-to-zero process to obtain few-step generative ability. Differently, existing fast sampling methods do not change the forward diffusion process but adopt high-order ODE solvers or distillation techniques for few-step generations. In fact, the latter can potentially be applied to DDMs.

Table 1: Unconditional generative performances (FID↓) on CIFAR-10 and CelebA-HQ-256 compared to previous DPMs including DDPM, SDE, LSGM, and CLD. Model size *w.r.t* the number of parameters is shown. 🟩 means FID is lower than the second best method with statistical significance (p-value $< 0.05$) based on the two sample t-test. 🟨 means the difference between our method and the closest best method is not statistical significant. We did not run the 2000-step sampling on CelebA-HQ-256, because it takes more than 7 days on an RTX 3090 GPU.

| Dataset | Method\NFE | Param | 10 | 20 | 50 | 1000 | 2000 |
|---|---|---|---|---|---|---|---|
| CIFAR-10 | DDPM | 143M | 296.84 | 140.68 | 38.36 | 3.17 | - |
| | SDE | 103M | 425.67 | 250.29 | 101.43 | 2.39 | 2.20 |
| | LSGM | 470M | 27.64 | 11.69 | 5.19 | 2.19 | - |
| | CLD | 103M | 415.36 | 162.44 | 52.70 | 2.27 | 2.23 |
| | DDM (ours) | 175M | **13.92**🟩$_{\pm0.054}$ | **8.92**🟩$_{\pm0.036}$ | 5.09🟨$_{\pm0.028}$ | 2.40 | 2.19 |
| CelebA-HQ-256 | SDE | 62M | 430.84 | 210.54 | 78.98 | 7.25 | - |
| | LSGM | 470M | 27.53 | 14.39 | 13.85 | 7.23 | - |
| | CLD | 103M | 355.15 | 179.85 | 64.77 | 8.06 | - |
| | DDM (ours) | 190M | 27.45🟨$_{\pm0.287}$ | 14.37🟨$_{\pm0.125}$ | **11.72**🟩$_{\pm0.103}$ | 7.10 | - |

## 5 EXPERIMENTS

### 5.1 EXPERIMENTAL SETUP

**Implementation Details.** We perform experiments on both unconditioned and conditioned image generation. For unconditional image generation, we evaluate DDM on the low-resolution CIFAR-10 dataset (Krizhevsky et al., 2009) as well as the high-resolution dataset CelebA-HQ-256 (Karras et al., 2018). For conditional image generation, we consider three tasks: image inpainting on CelebA-HQ-256, image super-resolution on DIV2K (Agustsson & Timofte, 2017), and saliency detection on DUTS (Wang et al., 2017). Due to the limitations of computing resources, we translate the diffusion process into latent space for high-resolution ($>$256) image synthesis following Rombach et al. (2022). We use the AdamW optimizer with scheduled learning rates and train DDM for 800k iterations (training details can be found in Appendix B). We follow Song et al. (2021c) using NCSNv2 (Song & Ermon, 2020) as the basic model. Differently, since DDM aims to predict the image and noise components separately, we modify the original U-Net architecture by adding an extra decoder. For conditioned image generation, we use an additional encoder to extract features for the conditional input which are used to correlate the features of $\mathbf{x}_t$ by attention mechanisms. For the selection of $\mathbf{h}_t$, we use a simple yet effective function $\mathbf{h}_t = \mathbf{c}$.

**Evaluation.** We use the Fréchet inception distance (FID) to measure image sample quality for unconditioned image generation. We generate 50,000 and 30,000 samples for CIFAR-10 and Celeb-A-HQ respectively to calculate FID. For super-resolution, we use the mean peak signal to noise ratio (PSNR) of the test set for evaluation. For image inpainting, we follow Suvorov et al. (2022) to use 26,000 images for training and 2,000 images for testing, and report FID metric. For saliency detection, we calculate the mean absolute error (MAE) on the whole test set. We also record the number of function evaluations (NFEs) during synthesis when comparing with previous DPMs.

### 5.2 UNCONDITIONED GENERATION

We present results and compare methods in Table 1. Two observations are made below.

**DDM is very competitive in long-step image generation.** On the CIFAR-10 and CelebA-HQ-256 datasets, DDM achieves FID of 2.19 and 7.10, respectively, using 2,000 and 1,000 function evaluations. This is very competitive compared with SDE, SLGM, CLD, and DDPM and demonstrates the validness of the DDM framework. Note that LSGM uses a much more lager model with 470M parameters while DDM only has 175M.

**DDM has superior performance in few-step generation.** On the CIFAR-10 dataset, DDM consistently outperforms existing methods under 10 and 20 steps. Using 10 steps, FID of our method is 13.72 lower than the second best method LSGM, despite the fact that LSGM is much larger in size. On CelebA-HQ-256, our method is also very competitive under all the few-step scenarios. Note that for fair comparison, we use Euler-Maruyama (Artemiev et al., 1995) to solve the competing

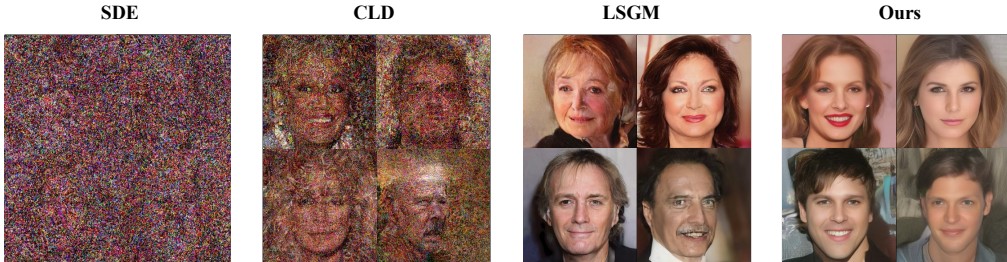

| SDE | CLD | LSGM | Ours |

Figure 4: Qualitative method comparison of unconditioned image generation with 10 function evaluations on CelebA-HQ-256. SDE, CLD, LSGM, and the proposed DDM are compared.

Table 2: Comparing DDM with DDPM and the state of the art on conditioned generation.

| Saliency detection | | Super-resolution | | Image inpainting | |
|---|---|---|---|---|---|
| Method | MAE↓ | Method | PSNR↑ | Method | FID↓ |
| Liu et al. (2023) | 0.026 | Ma et al. (2020) | 26.71 | Zhao et al. (2021) | 19.4 |
| Yun & Lin (2022) | 0.026 | Gao et al. (2023) | 27.10 | Zeng et al. (2022) | 7.28 |
| Lee et al. (2022) | 0.028 | Park et al. (2022) | 27.51 | Suvorov et al. (2022) | 6.13 |
| Lee et al. (2022) | **0.022** | Park et al. (2023) | 27.69 | Jeevan et al. (2023) | 5.59 |
| DDPM-5 step | 0.497 | DDPM-5 step | 9.13 | DDPM-5 step | 114 |
| DDPM-10 step | 0.473 | DDPM-10 step | 12.46 | DDPM-10 step | 95.8 |
| DDM-5 step (Ours) | 0.026 | DDM-5 step (Ours) | 27.61 | DDM-5 step (Ours) | 5.53 |
| DDM-10 step (Ours) | 0.025 | DDM-10 step (Ours) | **28.30** | DDM-10 step (Ours) | **5.25** |

diffusion models. More qualitative results are provided in Fig. 4, where our method is significantly better than SDE and CLD under 10 function evaluations and on par with LSGM.

## 5.3 CONDITIONED GENERATION

**Conditioned image generation with 5 or 10 steps.** We compare the proposed DDM with both DDPM and state-of-the-art methods for conditioned image generation. **(1) Saliency detection.** As shown in Tab. 2, DDM shows very superior performances to DDPM for few-step generation. Moreover, DDM achieves very competitive performance with state-of-the-art methods within 10-step generation. Note that TRACER-T7 gets the best result by using a much larger training size (640×640) than other comparisons (384×384). **(2) Super-resolution.** Compared to state-of-the-art methods, DDM only takes 5 NFEs to get very comparable performance, and achieves the best PSNR of 28.30 through 10 NFEs. **(3) Image inpainting.** We can observe that DDM has great advantages in image inpainting that it only uses 5 NFEs to exceed other state-of-the-art methods. The above experiments demonstrate the effectiveness of the proposed DDM and it can be easily applied to downstream image-conditioned tasks.

**Conditioned image generation with more steps.** As shown in Tab. 3, we evaluate the long-step performance on image-conditioned tasks. Different from unconditioned generation, the improvement of increasing the NFE is very slight for DDM.

Table 3: Comparing DDM and DDPM on conditioned generation with various steps. We report MAE ↓, PSNR ↑, and FID ↓ for saliency detection, super-resolution, and image inpainting, *resp.*

| Method\NFE | Saliency detection | | | | Super-resolution | | | | Image inpainting | | | |
|---|---|---|---|---|---|---|---|---|---|---|---|---|
| | 10 | 20 | 50 | 100 | 10 | 20 | 50 | 100 | 10 | 20 | 50 | 100 |
| DDPM | 0.4731 | 0.3846 | 0.3059 | 0.1985 | 12.46 | 13.58 | 14.29 | 15.91 | 95.8 | 89.9 | 71.6 | 64.3 |
| DDM | 0.0253 | 0.0251 | 0.0252 | 0.0250 | 28.30 | 28.29 | 28.35 | 28.33 | 5.25 | 5.19 | 5.15 | 5.15 |

Table 4: Comparing different choices of $\mathbf{h}_t$ on CIFAR-10. FID for unconditioned image generation is used as metric. We report FID under 10, 50, and 1,000 steps.

| $\mathbf{h}_t$ \NFE | 10 | 50 | 1000 |
|---|---|---|---|
| $\mathbf{c}$ | **13.92** | **5.09** | **2.40** |
| $\mathbf{a}t + \mathbf{c}$ | 15.17 | 9.69 | 3.15 |
| $\mathbf{a}t^2 + \mathbf{b}t + \mathbf{c}$ | 22.43 | 11.05 | 4.59 |
| $e^{\mathbf{a}t} + \mathbf{c}$ | 28.15 | 12.15 | 5.33 |
| $\sin \mathbf{a}t$ | 58.94 | 30.46 | - |

Table 5: Hyperparameter, architecture, and loss analysis. $t$ is defined in Eq. 6. FID for 10-step unconditioned generation on CIFAR-10 is reported.

| | $\lambda_1$ | $\lambda_2$ | 10-step FID |
|---|---|---|---|
| | 1 | 1 | 14.24 |
| Hyperparameter | $1/t$ | $1/(1-t)$ | 13.92 |
| | $e^t$ | $e^{1-t}$ | 13.95 |
| Architecure | U-Net+conv | | 15.13 |
| | U-Net+decoder | | 13.92 |
| Loss type | $\ell_1$ | | 14.98 |
| | $\ell_2$ | | 13.92 |

## 5.4 FURTHER ANALYSIS

**Comparing different choices of $\mathbf{h}_t$.** We conduct the analysis experiments on CIFAR-10 dataset. Eq. 6 shows that we can construct the forward process with arbitrary explicit functions which are obtained by solving $\mathbf{x}_0 + \int_0^1 \mathbf{h}_t dt = \mathbf{0}$. Here, we compare five functions listed in Table 4. The constant function has the lowest FID under the three NFEs. The reason might be that the constant function only has one learnable term $\mathbf{c}$, reducing the learning difficulty for the model. On the other hand, the constant function implies that the image component undergoes uniform attenuation during the forward process, providing a clearer and more concise learning pathway. The sine function limits the value range of $\mathbf{h}_t$ to $[-1, 1]$, which hinders the modeling of image-to-zero process.

**Hyperparameter sensitivity analysis.** Here we assess the weighting hyperparameters $\lambda_1$ and $\lambda_2$ defined in Section 4.2. We set the baseline weights to: $\lambda_1 = \lambda_2 = 1$. Since $\mathbf{x}_t$ is closer to $\epsilon$ when $t \to 1$, the second term in Eq. 10 is much smaller than the first term when $t \to 1$. The opposite phenomenon can be observed for the first term when $t \to 0$. Therefore, we empirically set two types of weighting selection to balance the two terms. As show in Table 5, the time-dynamic weights can slightly improve the baseline. It would be interesting to study this in future.

**Comparing variants of the U-Net architecture.** Because DDM predicts both the noise and $\mathbf{x}_0$, we need to modify the architecture so that the modified U-Net has two outputs. There are two variants: **(1)** we use an additional branch, which consists of two stacked 3x3 convolutional layers with stride 2 and padding 1, from the feature map before the output layer; **(2)** we add a new decoder branch that has the same architecture as the original decoder to obtain the another output. We named the two architectures U-Net+conv and U-Net+decoder. Table 5 shows U-Net+decoder performs better.

**Comparing $\ell_1$ and $\ell_2$ loss in Eq. 10.** We also analyze the effect of $\ell_1$ and $\ell_2$ loss for supervising the network. Results in Table 5 indicate that the $\ell_2$ loss is more suitable compared with the $\ell_1$ loss.

**Computational time cost.** We use a computing server with Intel Xeon Silver 4210R CPU and RTX3090 GPU. The training time on CIFAR-10 and Celeb-A-HQ is around 20 GPU days and 30 GPU days, respectively. The inference time to generate one image on CIFAR-10 and Celeb-A-HQ is 1.05 second and 1.98 second in 10-step setting, respectively.

## 6 CONCLUSION

We introduce a new diffusion method named decoupled diffusion models (DDMs) that improve the generative abilities of DPMs. Different from prior approaches that only have an image-to-noise process, we propose to decouple it into two relatively simpler processes: image to zero and zero to noise. For image to zero, we use an analytic transition function to model the gradient of the image component, resulting in a diffusion process that enables the model to sample with an arbitrary step size. For unconditioned and conditioned image generation tasks, we show that DDM has consistently competitive performance under both long-step and few-step setups.

**Future work and broader impact.** In principle, DDM can be combined with high-order ODE solvers to further improve the performance of few-step generation. Yet we have not derived the formula of high-order solvers. Like other deep generative models, DDM has the potential to transform many different fields and industries by generating new insights and positive social impact.

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

## A  PROOFS AND DERIVATIONS

### A.1  EQUIVALENCE BETWEEN DDM AND PREVIOUS DIFFUSION PROCESS (HO ET AL., 2020)

The previous study (Song et al., 2021c) has proved that the mapping from the image to noise can be formulated by a Stochastic Differential Equation (SDE):

$$d\mathbf{x}_t = f_t \mathbf{x}_t dt + g_t d\mathbf{w}_t, \quad \mathbf{x}_0 \sim q(\mathbf{x}_0), \tag{12}$$

where $f_t \mathbf{x}_t$ and $g_t$ represent the drift term and diffusion term of the Itô diffusion process (Itô et al., 1996). Our decoupled diffusion process is directly described by the Itô integral that can be formulated by:

$$\mathbf{x}_t = \mathbf{x}_0 + \int_0^t \mathbf{h}_t dt + \int_0^t d\mathbf{w}_t, \quad \mathbf{x}_0 \sim q(\mathbf{x}_0). \tag{13}$$

Taking the derivative of the above formula, we get the similar differential form to Eq. 12:

$$d\mathbf{x}_t = \mathbf{h}_t dt + d\mathbf{w}_t, \quad \mathbf{h}_t \sim \int_0^t \mathbf{h}_t dt + \mathbf{x}_0 = \mathbf{0}, \quad \mathbf{x}_0 \sim q(\mathbf{x}_0), \tag{14}$$

Comparing Eq. 12 and Eq. 14, we can prove that the two formulas are equal when $f_t \mathbf{x}_t = h_t$ and $g_t = 1$. Therefore, the proposed decoupled diffusion process is equivalent to previous diffusion processes described by Eq. 12. Actually, our decoupled diffusion formula is a special form of Eq. 12 that they both can describe the mapping from the image to noise. Differently, the decoupled process allows the model to learn the image and noise components independently. We split the complex mapping from the original image to noise into two relatively simpler processes, reducing the learning difficulty and enabling using a explicit transition probability to model the image component.

A.2   PROOF OF TRAINING OBJECTIVE

We can derive the training objective of DDM by maximizing the evidence lower bound of the log-likelihood $\log p(\mathbf{x}_0)$. Considering the continuous-time Markov chain, $\log p(\mathbf{x}_0)$ is represented by:

$$
\begin{aligned}
\log p(\mathbf{x}_0) &= \log \int p(\mathbf{x}_{0:1}) \mathrm{d}\mathbf{x}_{\Delta t:1}, \Delta t \to 0^+ \\
&= \log \int \frac{p(\mathbf{x}_{0:1})q(\mathbf{x}_{\Delta t:1}|\mathbf{x}_0)}{q(\mathbf{x}_{\Delta t:1}|\mathbf{x}_0)} \mathrm{d}\mathbf{x}_{\Delta t:1} \\
&= \log \mathbb{E}_q[\frac{p(\mathbf{x}_{0:1})}{q(\mathbf{x}_{\Delta t:1}|\mathbf{x}_0)}] \\
&\geq \mathbb{E}_q[\log \frac{p(\mathbf{x}_{0:1})}{q(\mathbf{x}_{\Delta t:1}|\mathbf{x}_0)}] \quad \#\text{Jensen's inequality} \\
&= \mathbb{E}_q[\log \frac{p(\mathbf{x}_1)\prod_{t=\Delta t}^{1} p_{\boldsymbol{\theta}}(\mathbf{x}_{t-\Delta t}|\mathbf{x}_t)}{\prod_{t=\Delta t}^{1} q(\mathbf{x}_t|\mathbf{x}_{t-\Delta t})}] \\
&= \mathbb{E}_q[\log \frac{p(\mathbf{x}_1)\prod_{t=\Delta t}^{1} p_{\boldsymbol{\theta}}(\mathbf{x}_{t-\Delta t}|\mathbf{x}_t)}{\prod_{t=\Delta t}^{1} q(\mathbf{x}_t|\mathbf{x}_{t-\Delta t}, \mathbf{x}_0)}] \quad \#\text{Markov property} \\
&= \mathbb{E}_q[\log \frac{p(\mathbf{x}_1)\prod_{t=\Delta t}^{1} p_{\boldsymbol{\theta}}(\mathbf{x}_{t-\Delta t}|\mathbf{x}_t)q(\mathbf{x}_{t-\Delta t}|\mathbf{x}_0)}{\prod_{t=\Delta t}^{1} q(\mathbf{x}_{t-\Delta t}|\mathbf{x}_t, \mathbf{x}_0)q(\mathbf{x}_t|\mathbf{x}_0)}] \quad \#\text{Bayes rule} \\
&= \mathbb{E}_q[\log \frac{p(\mathbf{x}_1)\prod_{t=\Delta t}^{1} p_{\boldsymbol{\theta}}(\mathbf{x}_{t-\Delta t}|\mathbf{x}_t)}{\prod_{t=\Delta t}^{1} q(\mathbf{x}_{t-\Delta t}|\mathbf{x}_t, \mathbf{x}_0)} + \log \frac{\prod_{t=\Delta t}^{1} q(\mathbf{x}_{t-\Delta t}|\mathbf{x}_0)}{\prod_{t=\Delta t}^{1} q(\mathbf{x}_t|\mathbf{x}_0)}] \\
&= \mathbb{E}_q[\log \frac{p(\mathbf{x}_1)\prod_{t=\Delta t}^{1} p_{\boldsymbol{\theta}}(\mathbf{x}_{t-\Delta t}|\mathbf{x}_t)}{\prod_{t=\Delta t}^{1} q(\mathbf{x}_{t-\Delta t}|\mathbf{x}_t, \mathbf{x}_0)} + \log \frac{1}{q(\mathbf{x}_1|\mathbf{x}_0)}] \\
&= \mathbb{E}_q[\log \frac{p(\mathbf{x}_1)}{q(\mathbf{x}_1|\mathbf{x}_0)} + \log \frac{\prod_{t=\Delta t}^{1} p_{\boldsymbol{\theta}}(\mathbf{x}_{t-\Delta t}|\mathbf{x}_t)}{\prod_{t=\Delta t}^{1} q(\mathbf{x}_{t-\Delta t}|\mathbf{x}_t, \mathbf{x}_0)}] \\
&= \mathbb{E}_q[\log \frac{p(\mathbf{x}_1)}{q(\mathbf{x}_1|\mathbf{x}_0)}] - \prod_{t=\Delta t}^{1} \mathbb{E}_q[D_{KL}(q(\mathbf{x}_{t-\Delta t}|\mathbf{x}_t, \mathbf{x}_0)||p_{\boldsymbol{\theta}}(\mathbf{x}_{t-\Delta t}|\mathbf{x}_t)))]
\end{aligned}
\tag{15}
$$

The first term in the equation can be interpreted as the prior matching term, which ensures that the original data can be transformed into the noise distribution when $t = 1$. It has been demonstrated in Section 3 of the main paper that our forward process satisfies this requirement, eliminating the need to construct a loss function for this term. On the other hand, the latter term represents the denoising matching term and its purpose is to ensure consistency in the distribution at $\mathbf{x}_t$ from both the forward and reversed processes. To achieve this, we aim to learn the desired denoising transition step $p_{\boldsymbol{\theta}}(\mathbf{x}_{t-\Delta t}|\mathbf{x}_t)$ as an approximation to the tractable ground-truth denoising transition step $q(\mathbf{x}_{t-\Delta t}|\mathbf{x}_t, \mathbf{x}_0)$. Minimizing this term entails achieving the closest possible match between the two denoising steps, as quantified by their KL Divergence. In fact, the ground-truth denoising transition step $q(\mathbf{x}_{t-\Delta t}|\mathbf{x}_t, \mathbf{x}_0)$ is still a normal distribution. Therefore, minimizing the KL Divergence is equivalent to minimizing the error between their mean and variance, which means we can use the MSE (mean squared error) function to formulate the training object.

We first prove the normality of $q(\mathbf{x}_{t-\Delta t}|\mathbf{x}_t, \mathbf{x}_0)$. Giving the forward transition probability $q(\mathbf{x}_t|\mathbf{x}_0) = \mathcal{N}(\mathbf{x}_t; \mathbf{x}_0 + \mathbf{H}_t, t\mathbf{I})$, we have:

$$
\begin{aligned}
q(\mathbf{x}_{t-\Delta t}|\mathbf{x}_t, \mathbf{x}_0) &= \frac{q(\mathbf{x}_t|\mathbf{x}_{t-\Delta t}, \mathbf{x}_0)q(\mathbf{x}_{t-\Delta t}|\mathbf{x}_0)}{q(\mathbf{x}_t|\mathbf{x}_0)} \\
&= \frac{q(\mathbf{x}_t|\mathbf{x}_{t-\Delta t})q(\mathbf{x}_{t-\Delta t}|\mathbf{x}_0)}{q(\mathbf{x}_t|\mathbf{x}_0)} \quad \#\text{Markov property} \\
&= \frac{q(\mathbf{x}_t|\mathbf{x}_{t-\Delta t})\mathcal{N}(\mathbf{x}_{t-\Delta t}; \mathbf{x}_0 + \mathbf{H}_{t-\Delta t}, (t-\Delta t)\mathbf{I})}{\mathcal{N}(\mathbf{x}_t; \mathbf{x}_0 + \mathbf{H}_t, t\mathbf{I})}.
\end{aligned}
\tag{16}
$$

Here, we can easily prove that $q(\mathbf{x}_t|\mathbf{x}_{t-\Delta t})$ is also a normal distribution by the following derivation:

$$
\begin{aligned}
\mathbf{x}_t &= \mathbf{x}_0 + \mathbf{H}_t + \sqrt{t}\boldsymbol{\epsilon}, \boldsymbol{\epsilon} \sim \mathcal{N}(\mathbf{0}, \mathbf{I}) \\
&= \mathbf{x}_0 + \mathbf{H}_t - \mathbf{H}_{t-\Delta t} + \mathbf{H}_{t-\Delta t} + \sqrt{t - \Delta t}\boldsymbol{\epsilon}_1 + \sqrt{\Delta t}\boldsymbol{\epsilon}_2, \boldsymbol{\epsilon}_1, \boldsymbol{\epsilon}_2 \sim \mathcal{N}(\mathbf{0}, \mathbf{I}) \\
&= \mathbf{x}_0 + \mathbf{H}_{t-\Delta t} + \sqrt{t - \Delta t}\boldsymbol{\epsilon}_1 + \mathbf{H}_t - \mathbf{H}_{t-\Delta t} + \sqrt{\Delta t}\boldsymbol{\epsilon}_2 \\
&= \mathbf{x}_{t-\Delta t} + \mathbf{H}_t - \mathbf{H}_{t-\Delta t} + \sqrt{\Delta t}\boldsymbol{\epsilon}_2. \quad \#\text{Definition of } q(\mathbf{x}_t|\mathbf{x}_0)
\end{aligned}
\tag{17}
$$

Thus, $q(\mathbf{x}_t|\mathbf{x}_{t-\Delta t})$ is the normal distribution with mean $\mathbf{x}_{t-\Delta t} + \mathbf{H}_t - \mathbf{H}_{t-\Delta t}$ and variance $\Delta t\mathbf{I}$:

$$
q(\mathbf{x}_t|\mathbf{x}_{t-\Delta t}) = \mathcal{N}(\mathbf{x}_t; \mathbf{x}_{t-\Delta t} + \mathbf{H}_t - \mathbf{H}_{t-\Delta t}, \Delta t\mathbf{I}).
\tag{18}
$$

Substituting Eq. 18 into Eq. 16, we have:

$$
\begin{aligned}
q(\mathbf{x}_{t-\Delta t}|\mathbf{x}_t, \mathbf{x}_0) &= \frac{q(\mathbf{x}_t|\mathbf{x}_{t-\Delta t})\mathcal{N}(\mathbf{x}_{t-\Delta t}; \mathbf{x}_0 + \mathbf{H}_{t-\Delta t}, (t - \Delta t)\mathbf{I})}{\mathcal{N}(\mathbf{x}_t; \mathbf{x}_0 + \mathbf{H}_t, t\mathbf{I})} \\
&\propto \exp\{-\frac{1}{2}[\frac{(\mathbf{x}_t - \mathbf{x}_{t-\Delta t} - \mathbf{H}_t + \mathbf{H}_{t-\Delta t})^2}{\Delta t} + \frac{(\mathbf{x}_{t-\Delta t} - \mathbf{x}_0 - \mathbf{H}_{t-\Delta t})^2}{t - \Delta t}]\} \\
&= \exp\{-\frac{1}{2}[\frac{t}{\Delta t(t - \Delta t)}\mathbf{x}_{t-\Delta t}^2 \\
&\quad + \frac{\Delta t\mathbf{x}_0 + (t - \Delta t)(\mathbf{x}_t - \mathbf{H}_t) + t\mathbf{H}_{t-\Delta t}}{\Delta t(t - \Delta t)}(2\mathbf{x}_{t-\Delta t}) + C]\},
\end{aligned}
\tag{19}
$$

where $C$ is the term not related to $\mathbf{x}_{t-\Delta t}$. Eq. 19 contains $\mathbf{x}_0$ which is known in the forward process, and we can directly get the prior $\mathbf{x}_0$ from the forward transition probability $q(\mathbf{x}_t|\mathbf{x}_0)$:

$$
\mathbf{x}_0 = \mathbf{x}_t - \mathbf{H}_t - \sqrt{t}\boldsymbol{\epsilon}.
\tag{20}
$$

Comparing Eq. 19 with the standard normal distribution and substituting Eq. 20 into Eq. 19, we can obtain the mean $\widetilde{\mathbf{u}}$ and variance $\widetilde{\sigma}^2$ of $q(\mathbf{x}_{t-\Delta t}|\mathbf{x}_t, \mathbf{x}_0)$:

$$
\begin{aligned}
\widetilde{\mathbf{u}} &= \mathbf{x}_t + \mathbf{H}_{t-\Delta t} - \mathbf{H}_t - \frac{\Delta t}{\sqrt{t}}\boldsymbol{\epsilon}, \\
\widetilde{\sigma}^2 &= \frac{\Delta t(t - \Delta t)}{t}.
\end{aligned}
\tag{21}
$$

Thus, the proof of $q(\mathbf{x}_{t-\Delta t}|\mathbf{x}_t, \mathbf{x}_0)$ is complete.

We use a neural network with parameters $\boldsymbol{\theta}$ to parameterize $p_{\boldsymbol{\theta}}(\mathbf{x}_{t-\Delta t}|\mathbf{x}_t)$. In practice, the variance $\widetilde{\sigma}^2$ only depends on $t$ and $\Delta t$ so we only need to parameterize the mean. From Eq. 15, we can optimize $\boldsymbol{\theta}$ by minimizing the KL Divergence:

$$
\begin{aligned}
&\min_{\boldsymbol{\theta}} D_{KL}(q(\mathbf{x}_{t-\Delta t}|\mathbf{x}_t, \mathbf{x}_0)||p_{\boldsymbol{\theta}}(\mathbf{x}_{t-\Delta t}|\mathbf{x}_t)) \\
&= \min_{\boldsymbol{\theta}} D_{KL}(\mathcal{N}(\mathbf{x}_{t-\Delta t}; \widetilde{\mathbf{u}}, \widetilde{\sigma}^2\mathbf{I})||\mathcal{N}(\mathbf{x}_{t-\Delta t}; \mathbf{u}_{\boldsymbol{\theta}}, \widetilde{\sigma}^2\mathbf{I})) \\
&= \min_{\boldsymbol{\theta}} \frac{1}{2}(\mathbf{u}_{\boldsymbol{\theta}} - \widetilde{\mathbf{u}})^T(\widetilde{\sigma}^2\mathbf{I})^{-1}(\mathbf{u}_{\boldsymbol{\theta}} - \widetilde{\mathbf{u}}) \\
&= \min_{\boldsymbol{\theta}} \frac{1}{2\widetilde{\sigma}^2}\|\mathbf{u}_{\boldsymbol{\theta}} - \widetilde{\mathbf{u}}\|^2.
\end{aligned}
\tag{22}
$$

As $\mathbf{u}_{\boldsymbol{\theta}}$ is also conditioned on $\mathbf{x}_t$, we can match $\widetilde{\mathbf{u}}$ closely by setting it to the following form:

$$
\mathbf{u}_{\boldsymbol{\theta}} = \mathbf{x}_t + \mathbf{H}_{\boldsymbol{\theta}t-\Delta t} - \mathbf{H}_{\boldsymbol{\theta}t} - \frac{\Delta t}{\sqrt{t}}\boldsymbol{\epsilon}_{\boldsymbol{\theta}}.
\tag{23}
$$

Then, the optimization problem simplifies to:

$$
\min_{\boldsymbol{\theta}} D_{KL}(q(\mathbf{x}_{t-\Delta t}|\mathbf{x}_t, \mathbf{x}_0)||p_{\boldsymbol{\theta}}(\mathbf{x}_{t-\Delta t}|\mathbf{x}_t))
$$

$$
= \min_{\boldsymbol{\theta}} \frac{1}{2\widetilde{\sigma}^2}\|(\mathbf{x}_t + \mathbf{H}_{\boldsymbol{\theta}\, t-\Delta t} - \mathbf{H}_{\boldsymbol{\theta}\, t} - \frac{\Delta t}{\sqrt{t}}\boldsymbol{\epsilon}_{\boldsymbol{\theta}}) - (\mathbf{x}_t + \mathbf{H}_{t-\Delta t} - \mathbf{H}_t - \frac{\Delta t}{\sqrt{t}}\boldsymbol{\epsilon})\|^2
$$

$$
= \min_{\boldsymbol{\theta}} \frac{1}{2\widetilde{\sigma}^2}\| \int_t^{t-\Delta t} \mathbf{h}_{\boldsymbol{\theta}\, t}\mathrm{d}t - \int_t^{t-\Delta t} \mathbf{h}_t\mathrm{d}t + \frac{\Delta t}{\sqrt{t}}(\boldsymbol{\epsilon}_{\boldsymbol{\theta}} - \boldsymbol{\epsilon})\|^2
$$

$$
= \min_{\boldsymbol{\theta}} \frac{1}{2\widetilde{\sigma}^2}\| \int_t^{t-\Delta t} \mathbf{h}_{\boldsymbol{\theta}\, t} - \mathbf{h}_t\mathrm{d}t + \frac{\Delta t}{\sqrt{t}}(\boldsymbol{\epsilon}_{\boldsymbol{\theta}} - \boldsymbol{\epsilon})\|^2 \tag{24}
$$

$$
=: \min_{\boldsymbol{\theta}} \frac{1}{2\widetilde{\sigma}^2}\|\mathbf{h}_{\boldsymbol{\theta}\, t} - \mathbf{h}_t\|^2 + \frac{1}{2\widetilde{\sigma}^2}\|\frac{\Delta t}{\sqrt{t}}(\boldsymbol{\epsilon}_{\boldsymbol{\theta}} - \boldsymbol{\epsilon})\|^2
$$

$$
= \min_{\boldsymbol{\theta}} \frac{1}{2\widetilde{\sigma}^2}\|\mathbf{h}_{\boldsymbol{\theta}\, t} - \mathbf{h}_t\|^2 + \frac{\Delta t^2}{2\widetilde{\sigma}^2 t}\|\boldsymbol{\epsilon}_{\boldsymbol{\theta}} - \boldsymbol{\epsilon}\|^2.
$$

As $\mathbf{h}_t$ depends on $\mathbf{h}_t$ and $\mathbf{h}_t$ is determined by its hyper-parameters $\phi$ that can be solved using $\mathbf{x}_0 + \int_0^1 \mathbf{h}_t\mathrm{d}t = \mathbf{0}$, we can directly parameterize $\phi$ using $\phi_{\boldsymbol{\theta}}$. Thus, the training objective is formulated by:

$$
\min_{\boldsymbol{\theta}} D_{KL}(q(\mathbf{x}_{t-\Delta t}|\mathbf{x}_t, \mathbf{x}_0)||p_{\boldsymbol{\theta}}(\mathbf{x}_{t-\Delta t}|\mathbf{x}_t))
$$

$$
= \min_{\boldsymbol{\theta}} \frac{1}{2\widetilde{\sigma}^2}\|\phi_{\boldsymbol{\theta}} - \phi\|^2 + \frac{\Delta t^2}{2\widetilde{\sigma}^2 t}\|\boldsymbol{\epsilon}_{\boldsymbol{\theta}} - \boldsymbol{\epsilon}\|^2. \tag{25}
$$

The proof of our training objective is completed. In essential, $\phi$ represents the original image component while $\boldsymbol{\epsilon}$ can be seen as the noise distribution. Therefore, the training objective allows the model to learn the image and the noise components independently.

## A.3 DERIVATION OF FEW-STEP SAMPLING FORMULA

Different from previous diffusion probabilistic models (DPMs), the proposed DDM naturally enables few-step sampling. Following Eq.17, we have:

$$
\mathbf{x}_t = \mathbf{x}_0 + \mathbf{H}_t + \sqrt{t}\boldsymbol{\epsilon}, \boldsymbol{\epsilon} \sim \mathcal{N}(\mathbf{0}, \mathbf{I})
$$

$$
= \mathbf{x}_0 + \mathbf{H}_t - \mathbf{H}_{t-s} + \mathbf{H}_{t-s} + \sqrt{t-s}\boldsymbol{\epsilon}_1 + \sqrt{s}\boldsymbol{\epsilon}_2, \boldsymbol{\epsilon}_1, \boldsymbol{\epsilon}_2 \sim \mathcal{N}(\mathbf{0}, \mathbf{I})
$$

$$
= \mathbf{x}_0 + \mathbf{H}_{t-s} + \sqrt{t-s}\boldsymbol{\epsilon}_1 + \mathbf{H}_t - \mathbf{H}_{t-s} + \sqrt{s}\boldsymbol{\epsilon}_2 \tag{26}
$$

$$
= \mathbf{x}_{t-s} + \mathbf{H}_t - \mathbf{H}_{t-s} + \sqrt{s}\boldsymbol{\epsilon}_2. \quad \#\text{Definition of } q(\mathbf{x}_t|\mathbf{x}_0)
$$

Thus, we can give the transition probability $q(\mathbf{x}_t|\mathbf{x}_{t-s})$ for an arbitrary step size $s$:

$$
q(\mathbf{x}_t|\mathbf{x}_{t-s}) = \mathcal{N}(\mathbf{x}_t; \mathbf{x}_{t-s} + \mathbf{H}_t - \mathbf{H}_{t-s}, s\mathbf{I}). \tag{27}
$$

With Eq. 27, we can easily derive the mean and variance of $q(\mathbf{x}_{t-s}|\mathbf{x}_t, \mathbf{x}_0)$ following Eq. 19:

$$
q(\mathbf{x}_{t-s}|\mathbf{x}_t, \mathbf{x}_0) \propto \exp\{-\frac{(\mathbf{x}_{t-s} - \widetilde{\mathbf{u}})^2}{2\widetilde{\sigma}^2\mathbf{I}}\},
$$

$$
\widetilde{\mathbf{u}} = \mathbf{x}_t + \mathbf{H}_{t-s} - \mathbf{H}_t - \frac{s}{\sqrt{t}}\boldsymbol{\epsilon}, \tag{28}
$$

$$
\widetilde{\sigma}^2 = \frac{s(t-s)}{t}.
$$

Eq. 28 means DDM can sample with any step sizes even using the largest step size $s = 1$ for one-step generation. However, it is challenging to estimate an exactly accurate $\phi$ at the initial time, and one-step generation causes the variance $\widetilde{\sigma}^2$ to be zero. This results in the generated images appearing somewhat blurry and lacking diversity. Therefore, we still sample iteratively but use a much larger step size than previous DPMs, reducing the sampling steps from 1000 to 10 drastically. We commence at $t = 1$ and take uniform steps of size $s$ until we reach $t = 0$.

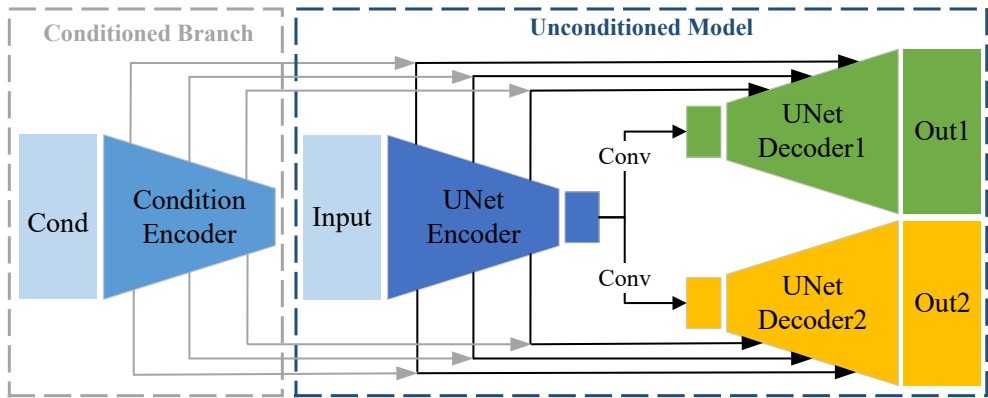

Figure 5: Architecture detail. The 'Cond' represents the conditioned input. The conditional branch is only used in conditioned generation tasks.

## B   MORE DETAILS OF EXPERIMENTS

### B.1   DETAILS OF PRELIMINARY EXPERIMENTS.

In the preliminary experiments, we use a small U-Net architecture with the feature channel 64 and channel multiplier [1, 2, 4, 8]. We only train all preliminary models for 200k iterations. For DDM model used in the preliminary experiments, we add an additional decoder based on the small U-Net and formulate $\mathbf{h}_t = \mathbf{c}$.

### B.2   ARCHITECTURE OF NETWORK

As shown in Fig. 5, we modify the original U-Net architecture (Song & Ermon, 2020) and add an extra decoder so that our model has two outputs for predicting image and noise components respectively. For conditional generation tasks, we utilize a down-sampling encoder like the U-Net encoder to extract multi-level features of the conditional input, and concatenate these features and the image features with the same levels as the decoder's inputs. In practice, we use the Swin-B (Liu et al., 2021) as our condition encoder.

### B.3   TRAINING DETAILS

**Hyper-parameters**.   We provide an overview of the hyper-parameters of all trained DDMs in Tab. 6. Different from previous models that usually adopt a constant learning rate, we implement the polynomial policy to decay the learning rate gradually, which can be formulated by: $\gamma = \max(\gamma_0 \cdot (1 - N_{iter}/Ntotal)^p, \gamma_{min})$. Here $\gamma_0$ is the initial learning rate and $\gamma_{min}$ denotes the smallest learning rate, $N_{iter}$ and $N_{total}$ correspond to the current iteration number and total iteration number, $p$ a hyper-parameter and we set it to 0.96. Additionally, we employ the exponential moving average (EMA) to prevent unstable model performances during the training process. We have observed that using mixed-precision (FP16) training negatively impacts the generative performances, hence, we do not utilize it.

**Obtaining ground truth of $\phi_{\boldsymbol{\theta}}$**.   In the training phase, we obtain the ground truth by solving $\mathbf{x}_0 + \int_0^1 \mathbf{h}_t \mathrm{d}t = \mathbf{0}$. For a simple example $\mathbf{h}_t = \mathbf{c}$, the only parameter of $\mathbf{h}_t$ is $\mathbf{c}$ and we can easily get: $\mathbf{c} = -\mathbf{x}_0$. Thus, the ground truth of $\phi_{\boldsymbol{\theta}}$ is $-\mathbf{x}_0$. For the linear function $\mathbf{h}_t = \mathbf{a}t + \mathbf{c}$, we can not solve the two parameters $\mathbf{a}, \mathbf{c}$ using one equation. To avoid this problem, we first sample one of parameters from $\mathcal{N}(\mathbf{0}, \mathbf{I})$, and substitute it into $\mathbf{x}_0 + \int_0^1 \mathbf{h}_t \mathrm{d}t = \mathbf{0}$ to solve another parameter. In this way, we concatenate $\mathbf{a}, \mathbf{b}$ as the ground truth of $\phi_{\boldsymbol{\theta}}$. The ground truths of other functions can be solved in a similar way.

Table 6: Hyper-parameters for the trained DDMs.

| Task | Unconditional generation | | Inpainting | Super Resolution | Saliency Detection |
|---|---|---|---|---|---|
| | CIFAR10 | CelebA-HQ-256 | | | |
| Image size | 32×32 | 256×256 | 256×256 | 512×512 | 384× 384 |
| Batch size | 128 | 48 | 48 | 12 | 16 |
| Learning rate | 1e-4∼1e-5 | 5e-5∼5e-6 | 4e-5∼4e-6 | 5e-5∼5e-6 | 5e-5∼5e-6 |
| Iterations | 800k | 800k | 400k | 400k | 400k |
| Feature channels | 192 | 96 | 96 | 128 | 128 |
| Channel multiplier | [1, 2, 2, 2] | [1, 2, 3, 4] | [1, 2, 4, 8] | [1, 2, 4, 4] | [1, 2, 4, 4] |
| Number of blocks | 3 | 3 | 2 | 2 | 2 |
| Smallest time step | 1e-4 | 1e-4 | 1e-4 | 1e-4 | 1e-4 |

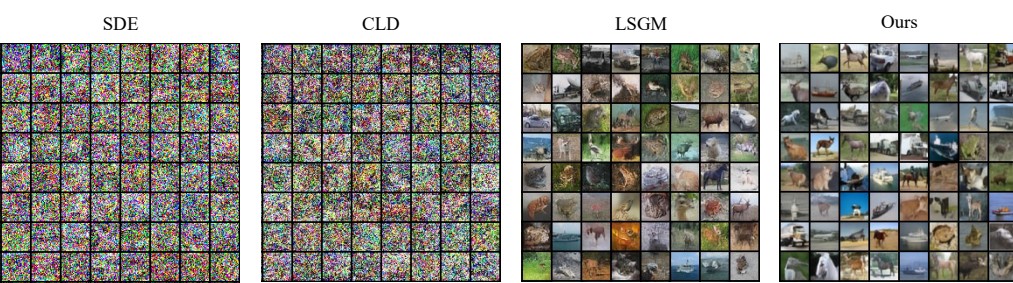

Figure 6: Comparisons of 10-step unconditional generation on CIFAR10.

**Final denoising step.** In general, the generated samples typically contain small noise that is hard to detect by humans (Jolicoeur-Martineau et al., 2021). To remove this noise, we follow Song et al. (2021c) letting the last denoising step occur at $t = \Delta t$ where $\Delta t$ is the smallest step size.

## C  ADDITIONAL VISUAL RESULTS

We present more visual comparisons in the following figures: Fig. 6 shows the visual comparisons between DDM and other DPMs on CIFAR10 dataset, and Fig. 7 shows the visual comparisons on CelebA-HQ-256 dataset. Additionally, we show more visual results of both unconditional and conditional generation tasks in Fig. 8- 12, which demonstrates our method can generate high-quality images only using 10 function evaluations.

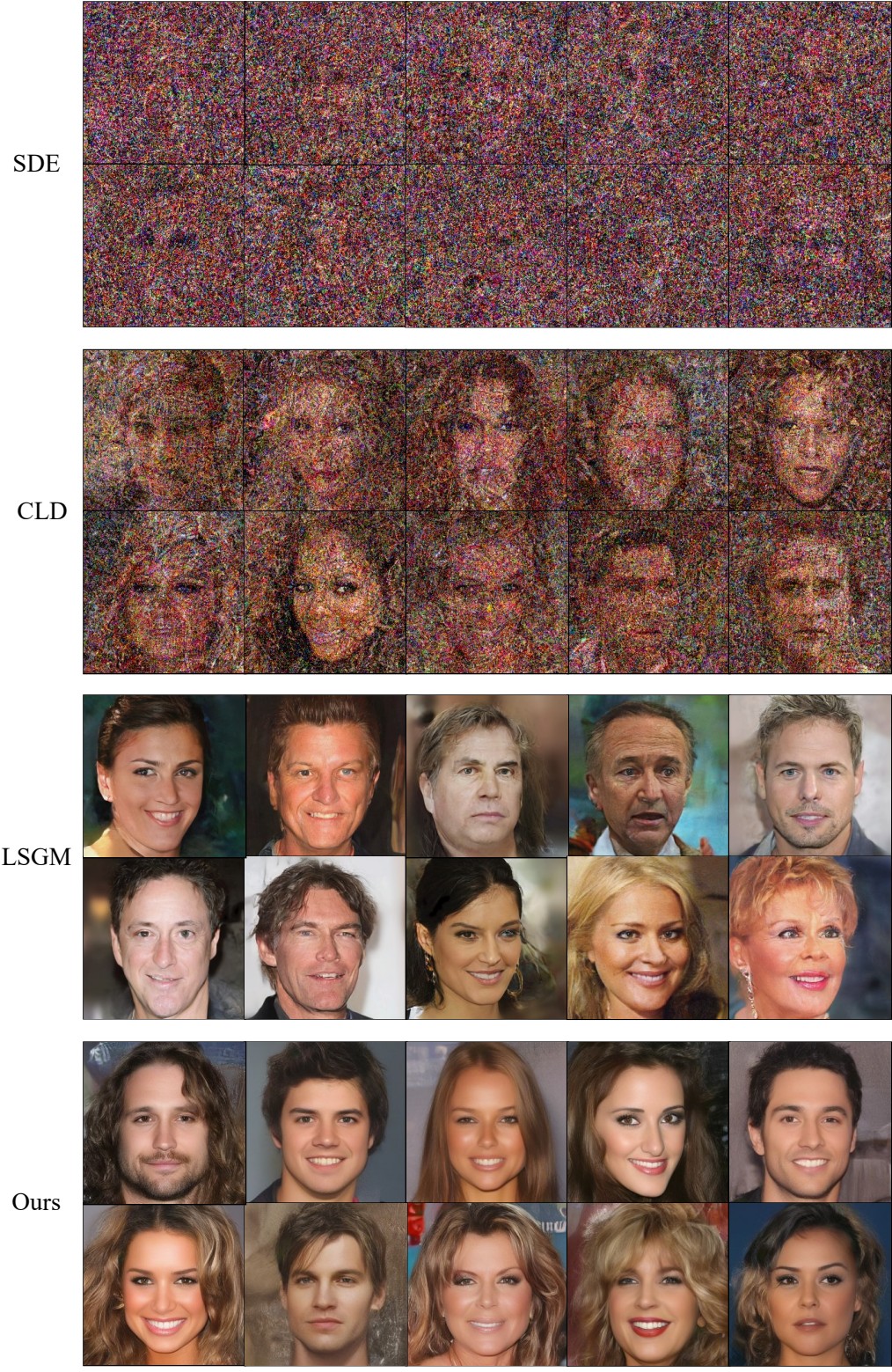

Figure 7: Comparisons of 10-step unconditional generation on CelebA-HQ-256.

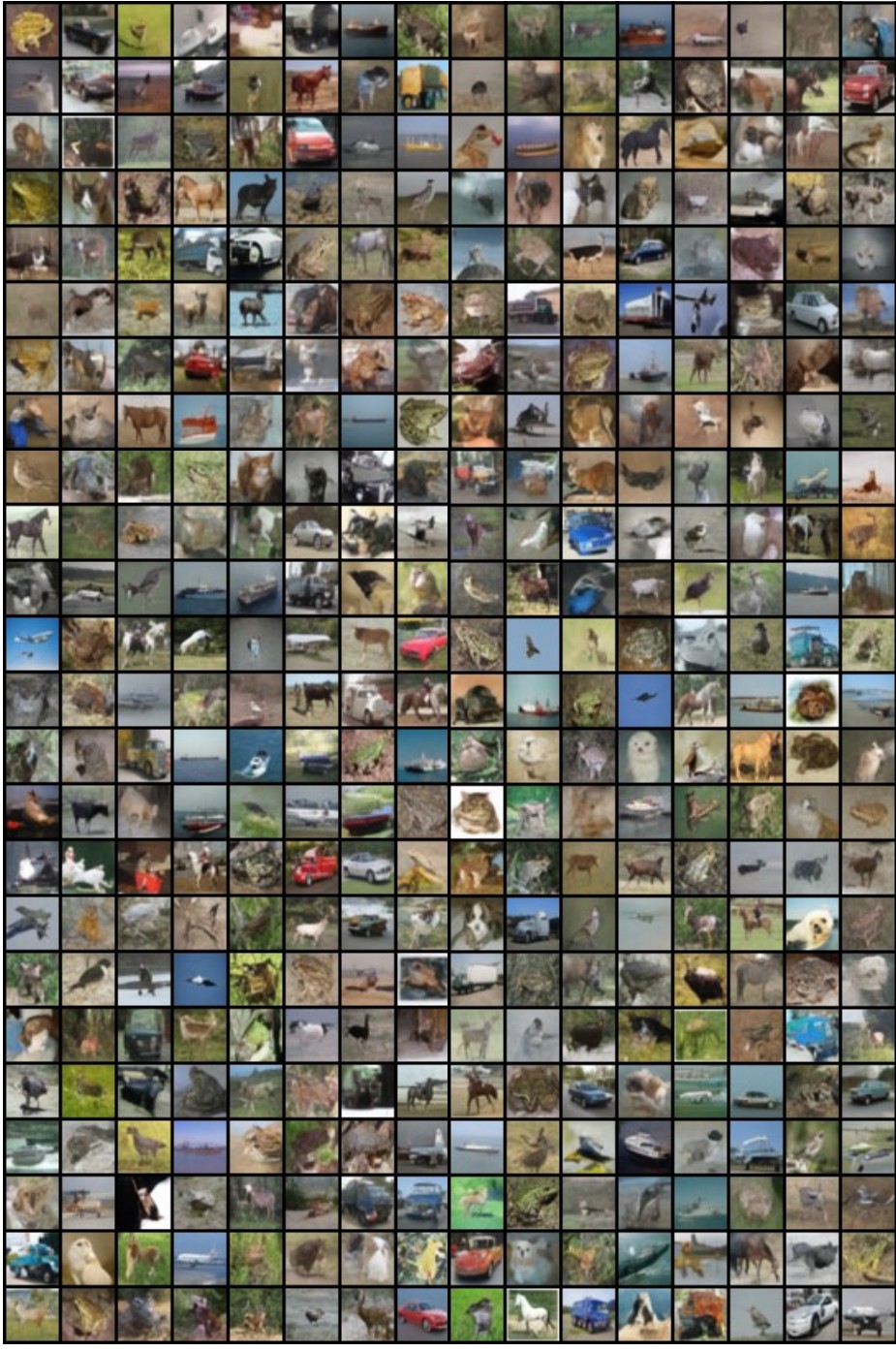

Figure 8: 10-step unconditional generation on CIFAR-10.

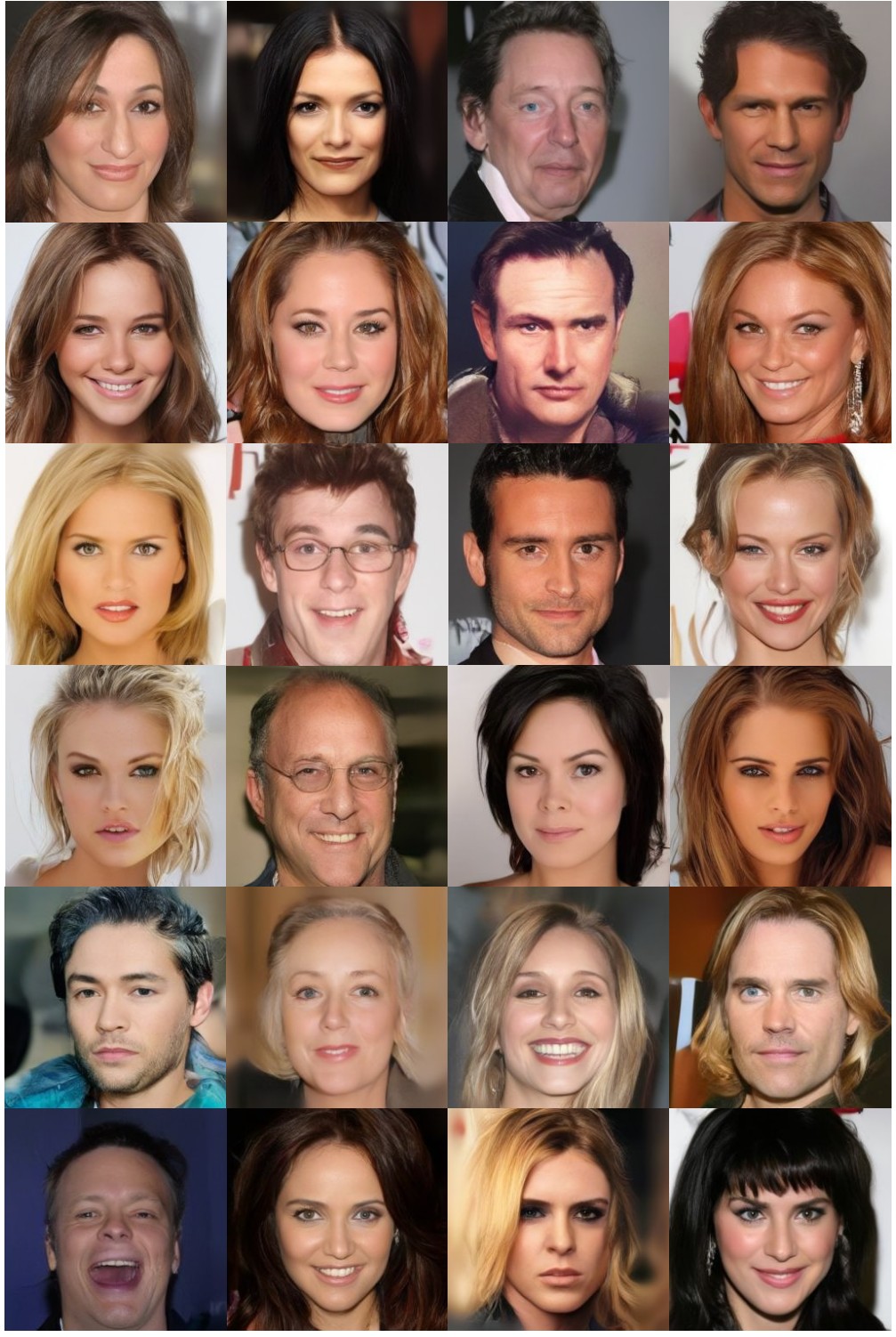

Figure 9: 10-step unconditional generation on CelebA-HQ-256.

Input

Generation

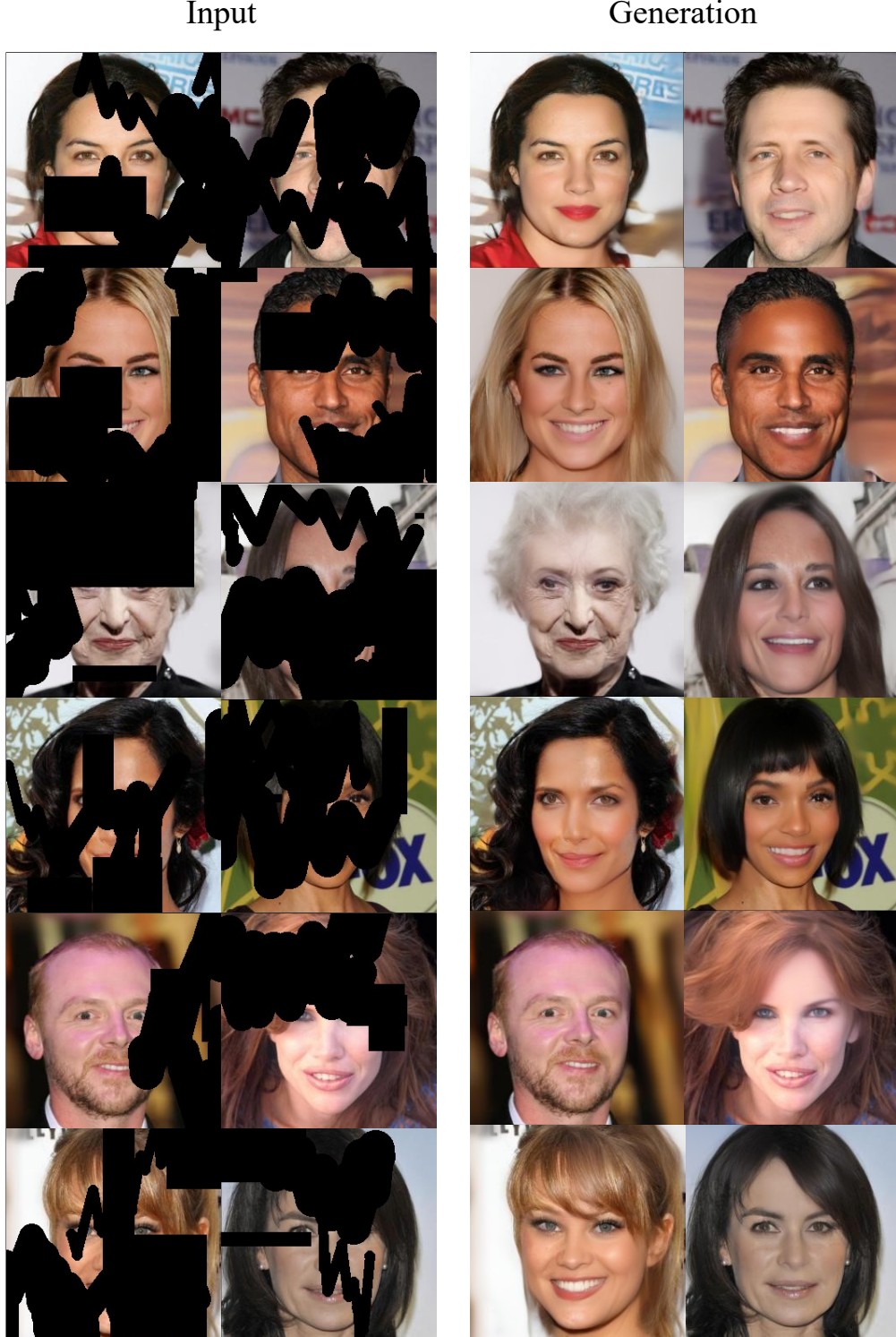

Figure 10: 10-step inpainting visualization.

| Input | Generation | Input | Generation |
|-------|-----------|-------|-----------|

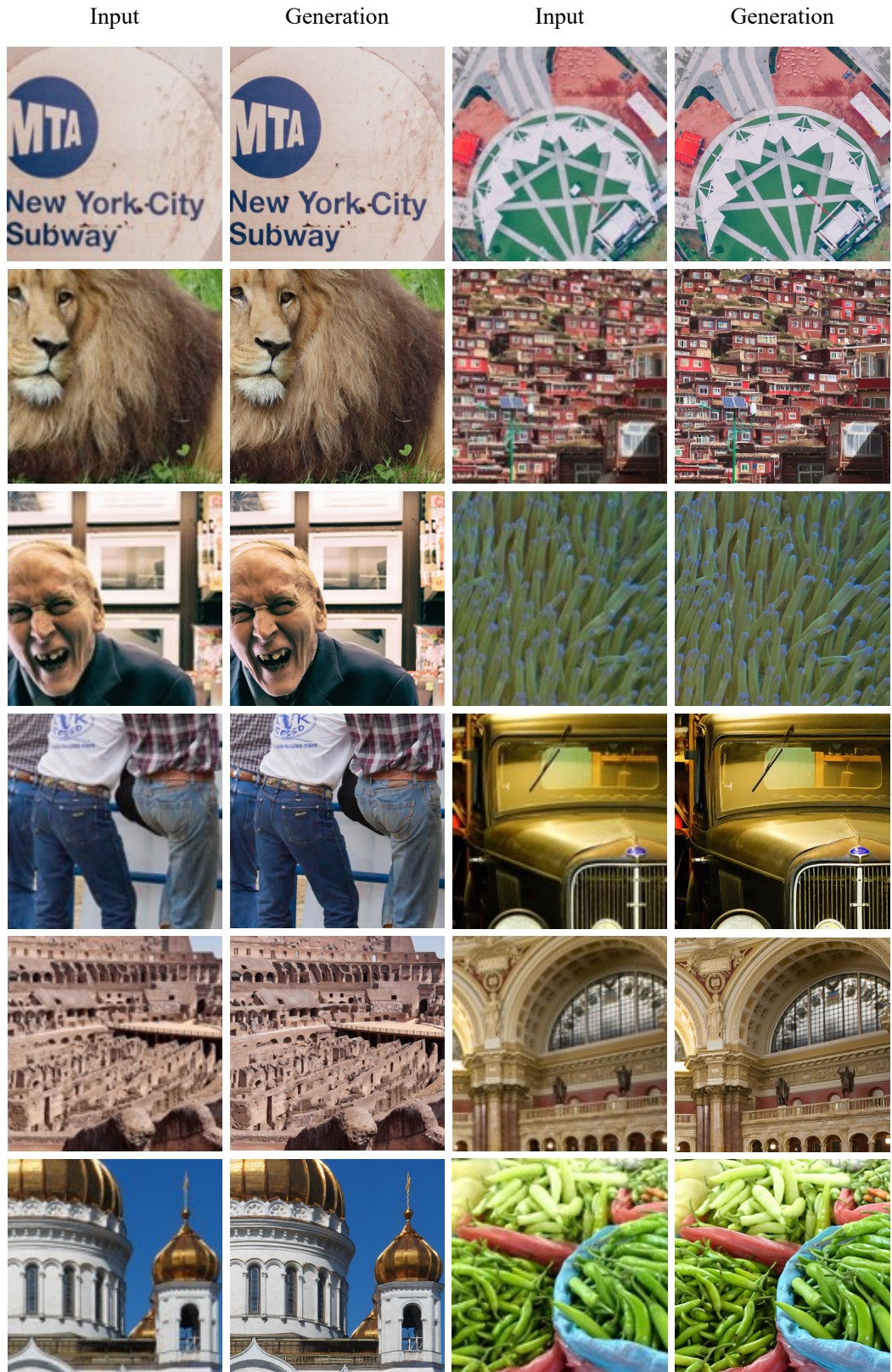

Figure 11: 10-step super-resolution visualization.

Input          Generation          Input          Generation

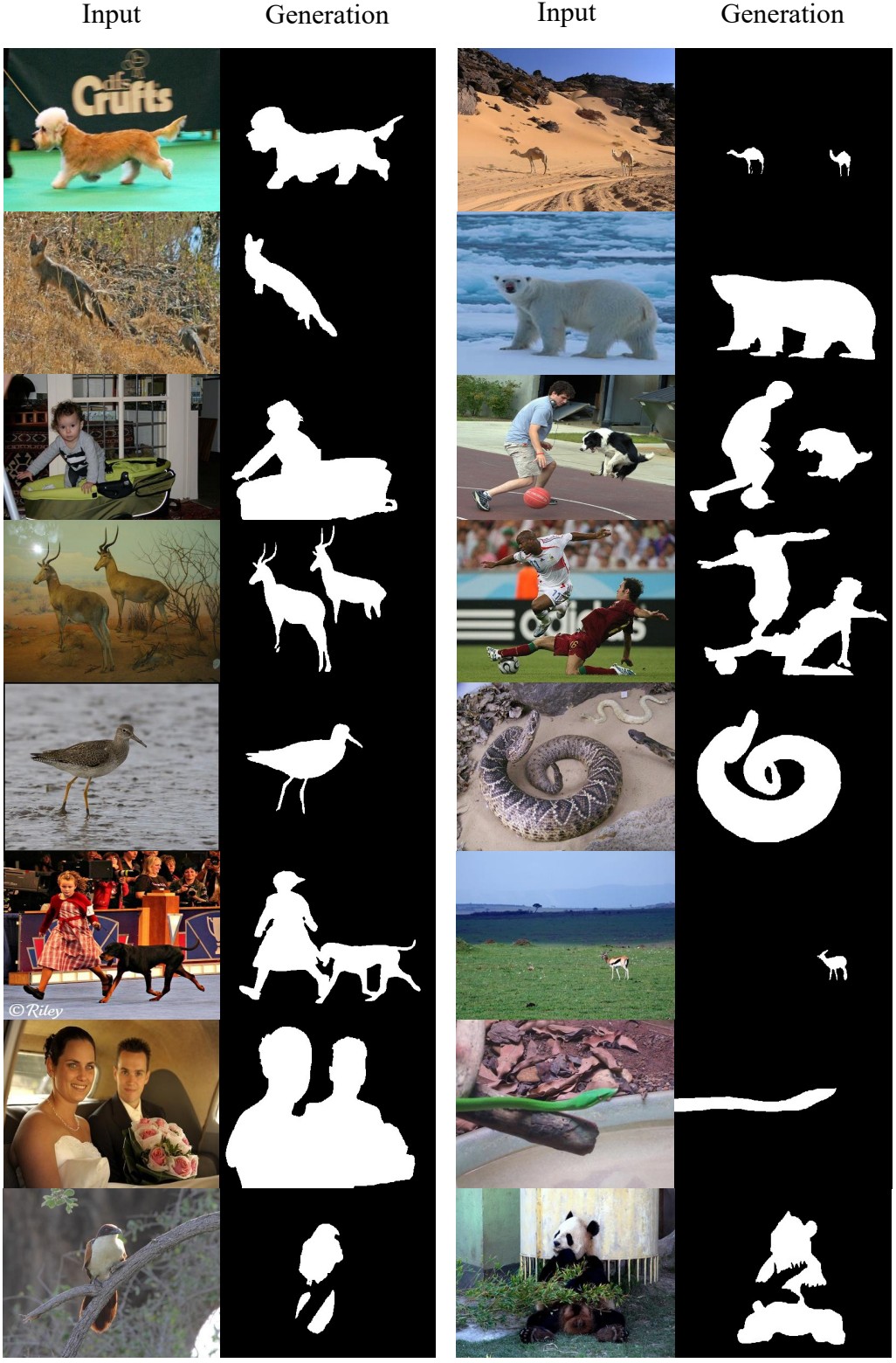

Figure 12: 10-step saliency detection visualization.

