# OpenReview forum: "Decoupled Diffusion Models: Image to Zero and Zero to Noise"
_ICLR.cc/2024/Conference — ICLR 2024 Conference Withdrawn Submission_

### Official Review · Reviewer_7ZTA · 2023-10-23

**Soundness:** 2 fair
**Presentation:** 2 fair
**Contribution:** 1 poor
**Rating:** 3
**Confidence:** 5

**Summary:**

The paper proposes “Decoupled Diffusion Models (DDM)”, which is a new forward/backward SDE for diffusion and corresponding learning objective. The authors claimed the new SDE to have de-coupling characteristics, i.e. it “decoupled” the data-to-zero and zero-to-noise. Authors claim that due to this de-coupled nature, it’s easy to train and also “naturally perform sampling with arbitrary step size”. The paper shows, in experiment section, unconditional and conditional evaluations of the DDM comparing with competing methods proving it to be competitive and sometimes superior.

**Strengths:**

The overall idea, the way it is presented, seems new. There is a conceptual contribution of a new diffusion SDE, which might be interesting for others. Also the concept of “de-coupling” the two processes might be useful for future works to analyze.

**Weaknesses:**

Despite its apparent novel outlook, the paper has a lot of flaws — in terms of presentation, conceptual clarity and technical novelty. From a bird’s eye view, the idea seems novel and interesting (which I thought after reading the abstract). However, the flaws were visible after digging into into it a bit more. Also I had a little difficult time deciphering the paper due to its not-so-good writing quality and math notations (more on this later).

**Conceptual clarity and Technical novelty:**

- The core idea revolves around the new forward process in Eq. 6 of the paper, which looks different from that of traditional DDPM. But I don’t necessarily see how it is any novel, apart from it being a special case of traditional DDPM. It is not clearly shown in the paper, but here’s how: From your choice of $\mathbf{h}_t = -x_0$ (in Appendix B.3 second para) and $\int_0^td\mathbf{w}_t \sim \mathcal{N}(\mathbf{0}, t \cdot \mathbf{I})$, one can easily show Eq.6 to be

    $$
    x_t = x_0 + (-x_0)\int_0^t dt + \sqrt{t}\cdot \epsilon \\ = x_0 - x_0 \cdot t + \sqrt{t}\cdot \epsilon \\ = (1-t)\cdot x_0 + \sqrt{t}\cdot\epsilon
    $$

    This is effectively just a standard DDPM with a different noising schedule, i.e. $(\alpha_t = 1-t, \beta_t = \sqrt{t})$ in your notation. Different choices of $\mathbf{h}_t$ (in table 4) would also yield different noising schedule of standard DDPM.

- Even if, for a moment, we consider DDM to be indeed different, I still don’t see what exactly is de-coupled here. It seems the author’s concept of “de-coupling” is based on the premise that prior works (DDPM essentially) have something *coupled*. No its not — the noise $\hat{\epsilon} = \epsilon_{\theta}(\cdot)$ and image predictions $\hat{x_0} = x_{\theta}(\cdot)$ are *exactly* related to each other given $x_t$. But in practice, only one of them is used in most cases, not both. I would assume that two quantities are called “coupled” if they are independent and hence can’t be trivially separated. So, from this perspective, I don’t see any point of “de-coupling” them.
- For argument’s sake, even if you refute the above point, I still don’t see what exactly is decoupled in Eq. 6 ? It is still a coupled process where image-to-zero and zero-to-noise is added together at every step. By this logic, standard DDPM ($x_t = \alpha_t x_0 + \beta_t \epsilon$) is also de-coupled ? The only reason Eq. 6 seems to have some sort of de-coupling is because of its attenuation process being *additive instead of multiplicative*. But then again, I showed above that’s it’s effectively multiplicative.
- Regarding the reverse process, the authors credited (4th paragraph in introduction) the “arbitrary step jump” property to the *analytic nature* of the zero-to-image process (line 4 of algo 2). However, one must realize that this maybe analytic, but not error-free. The authors used quite a sloppy notation which made it hard to see it clearly. If you use hats (differentiaing from line 4 of algo 1) for predicted quantities, you see that $\hat{\mathbf{H}}_t = \int_0^t \mathbf{h}^{\hat{\phi}} dt = -\int_0^t \hat{x}_0 dt$ incurs error too. So I don’t understand how this is a reasonable explanation. In fact, any DPM can perform arbitrary step sampling — how is this any new? Moreover, contrary to the standard DDPM, you now have two predicted quantity and hence two possible sources of error !

**Experiments and Evaluations:**

- I don’t fully understand what is being claimed to be superior — training or sampling performance, or both ? I assume its training, since you used the naive Euler-Maruyama sampler for everything (page 7 last line), which by the way, is already a bit worrying since there are far better samplers now.
- Table 1 results (prior works) are all with stochastic or determinitic (prob. flow ODE) sampling ? There aren’t much clear experimental details for the experiments to be conclusive.
- For table 1, can you please mention exactly which models have been used ? There is no citation on the prior works. E.g. DDPM/SDE model-size and FID numbers seem inconsistent from what I have personally worked with. Please see the EDM paper by Karras et al. for a comprehensive FID numbers.
- Figure 3 equates MSE loss with “image quality” ? Also, how did you measure the MSE error, as in where did you get the target GT ? It is an unconditional case where you cannot know before hand which sample it will generate, right ?

**Writing quality:**

- The overall quality of writing is just okay-ish, not too good.
    - In abstract: “we mathematically derive .. 2) for derive the sampling ..”
    - In still don’t understand what is an “analytic transition function” mentioned several times ?
    - Para 4 in Introduction: “In Fig. 4, either ..” — this seems to be a wrong reference.
    - Why did section 4.1 end up in the beginning of the method section ? Doesn’t make much sense — it’s full of implementation details which usually go in experiments.
- Related to math notations
    - Just after eq. 1: “$\beta_t$ is designed to increase ..” — there is another restriction of being in $[0, 1]$
    - Eq. 9 should write $\tilde{\sigma_t}^2$ instead of $\tilde{\sigma}^2$
    - I am still confused about $\phi$ and $\mathbf{c}$ notations with respect to $\mathbf{h}_t$. The notations here can be optimized a bit.
    - Why not write $\mathbf{h}_t^{\phi}$ instead of just $\mathbf{h}_t$ to emphasize that it’s a parametric function?
    - I already mentioned the problem of not using $\hat{\ }$ for predicted quantities.
    - Why not mention the arguments of functions, like $\phi_{\theta}(x_t, t)$ and $\epsilon_{\theta}(x_t, t)$ in Eq. 10 and elsewhere ? Also, Eq. 10 is missing the HPs $(\lambda_1, \lambda_2)$.

**Questions:**

Please see the weakness section for questions along with comments and suggestions.

---

### Official Review · Reviewer_UWnj · 2023-10-30

**Soundness:** 3 good
**Presentation:** 3 good
**Contribution:** 2 fair
**Rating:** 3
**Confidence:** 3

**Summary:**

This paper proposes decoupled diffusion models (DDMs) that aim to decouple the forward image-to-noise mapping into image-to-zero mapping and zero-to-noise mapping. Based on this idea, the training objective function and reverse sampling process are presented. The experiments show that DDM can produce visually high-quality images in less than 10 function evaluations.

**Strengths:**

* The presented DDM seems to be a new diffusion process in DPM literature.
* Both training and sampling processes are clearly presented in techniques, making the approach easy to follow.
* Experiments on several image tasks are conducted.
* The paper is well written.

**Weaknesses:**

* This paper aims to split the DPMs into an image-to-zero mapping and a zero-to-noise mapping. However, from Eq. 6 or Table 1, $\mathbf{x}_0+\int_0^t\mathbf{h}_t{\rm d}t$ is not zero, though the signal is degenerated. I suspect that the "image-to-zero mapping" may mean $\mathbf{x}_0+\int_0^1\mathbf{h}_t{\rm d}t=0$. However, in the DDPM or VP-SDE, the drift term is $\mathbf{x}_0e^{-\frac{1}{2}\int_0^T\beta(s){\rm d}s}$ also approaches zero as long as  $T$ is large. In conclusion, the paper is somewhat overclaimed, and meanwhile, the novelty is limited to some extent.
* On the reverse process, why approximate  $q(\\mathbf{x}\_{t-\\Delta t}|\\mathbf{x}_t,\\mathbf{x}_0) $ by $q(\\mathbf{x}\_{t-\\Delta t}|\\mathbf{x}_t).$ To my understanding, the approximation is reasonable only for small $\Delta t$. However, in the sampling process, to decrease sampling steps, $\Delta t$ should be not small. How about directly using the reverse SDE corresponding to Eq. 6 to develop the reverse sampling process?
* In Eq. 9, $\bf \epsilon\sim \mathcal{N}(0,I)$ is a noise sampled from Gaussian distribtion. Why do the authors use a network $\bf \epsilon_{\theta}$ to learn it?
* $\mathbf{H}_t$ has a analytical formulation. Why do the authors still use a network to predict its parameters rather than just use the analytical formulation?
* It's OK that the first term in Eq. 9 is motivated by section 4.1. But it is not mathematically deduced.
* How about the performance of DDM if we replace the first term in Eq. 9 as predicting $\mathbf{x}_0$? To show the effectiveness of DDM, such an ablation should be included.
* It is unclear why DDM can speed up the sampling process.
* The compared methods are limited, and some SOTA approaches, e.g., "Elucidating the Design Space of DBGM", are missing.
* The generated images by DDM seem to be too "smooth" and hence are not realistic.
* Due to the abovementioned issues on both novelty, claims, and experiments, the reviewer suggests the authors improve the paper, and then resubmit it to a future conference.

**Questions:**

See the weakness.

---

### Official Review · Reviewer_Jv4X · 2023-11-04

**Soundness:** 3 good
**Presentation:** 2 fair
**Contribution:** 2 fair
**Rating:** 5
**Confidence:** 4

**Summary:**

This paper proposed a decoupled diffusion model (DDM) by learning the image attenuation process and diffusion process jointly.
The image attenuation process can be constructed in multiple ways. A key observation is to solve a group of constraints.
This work derived an analytical framework for this join process, which in turn offers the reverse transition distribution for sampling.
The objective function could be split into two targets for predicting the attenuation process and noises using the analytical framework.

To back up the empirical benefits of the decoupling framework, this paper conducted unconditional and conditional image generation tasks.
Specifically, DDMs predict the attenuation process and noises together using a two-branch UNet decoder.
Experiments validated that the decoupled framework accelerated the sampling process to around 10 NFEs, showing advantages over previous continuous-time diffusion models and latent diffusion models.

**Strengths:**

- Analytic framework for the proposed join process. This backs up the proposed method and provides the sampling algorithm for the reverse time process.
- Detailed experiments. For the design choices in the image attenuation process and other hyperparameters, this work provides a variety of comparisons.
- Empirical benefits. The decoupled design offers sampling accelerations compared with LSGM and CLD.

**Weaknesses:**

I feel this work is overall novel concerning the technical content. However, I'd challenge the motivation of this work. In my view, despite the technical contributions, the analytical framework is more like the post hoc fashion "validation" of a simple experimental observation of predicting the attenuation jointly. This paper constructed a formal introduction to this method using the narrative and derivation. However, this is not a fundamental understanding or theory on why acceleration can be achieved by decoupling. It seems to me that cannot be understood as minimizing the transport cost (finding more straight paths), either.

That being said, although this work derived the analytical framework, it is not motivated in a principled manner or clearly **explained** using theory or experiments.

**Questions:**

There are several technical questions for the manuscript.

- Is the diffusion process a linear scheduler to a uniform variable? There is 't ∼ Uniform(0, 1)` in the algorithm box. It remains unclear to me if the linear scheduler is applied. The influence of the diffusion schedulers is also vague compared to the baseline models, as they may use different diffusion designs.
- The joint prediction design actually enlarges the model sizes. As a result, this work compares with the latent diffusion models like LSGM. LSGM is, for sure, a strong baseline. But I cannot ensure this is a fair comparison. There are also more compact models like EDM, achieving faster sampling under smaller model sizes. In addition, why are the model sizes of DDPM and SDE larger than their original papers'?
- What if only run 5 NFES?

---

### Official Review · Reviewer_D5Rq · 2023-11-07

**Soundness:** 2 fair
**Presentation:** 2 fair
**Contribution:** 2 fair
**Rating:** 3
**Confidence:** 4

**Summary:**

This work proposes an image generation model named decoupled diffusion model (DDM), with the intuition of decoupling the forward process of diffusion models into an “image-to-zero” mapping and a “zero-to-noise” mapping. That is, in the forward process, the clean image will be attenuated to zero and the added noise will grow to a standard Gaussian noise. With this intuition, this work introduces a new training objective that predicts noise and clean image, simultaneously. In the tasks of unconditional and image-conditional image synthesis, this work demonstrates that the proposed method can achieve better performance than SOTA methods when the number of function evaluations is small ($\leq$10 steps).

**Strengths:**

- Overall, the writing is clear, although there are some minor presentation issues (see comments below).
- Experimental settings are relatively diverse: Both unconditional and image-conditional image synthesis tasks are considered. Besides, the ablation studies are sufficiently conducted to show the importance of each component or hyperparameter.

**Weaknesses:**

- Regarding the new diffusion formula, I do not see much difference with VP-SDE except for the different noise scheduling. Considering the best performing case where $h_t =c$, from $x_0 + \int_0^1h_t dt =0$ we know $h_t = -x_0$. That is, the forward SDE becomes $dx = -x_0dt + dw$ and the analytic form of $x_t$ becomes $x_t = (1-t)x_0 + \sqrt{t}\epsilon$. Compared with VP-SDE where $x_t = s(t) x_0 + \sigma(t) s(t) \epsilon$ (note that here I followed the notations in Karras’ EDM paper: https://arxiv.org/abs/2206.00364), we can see the main difference is how to set the noise scheduling. For example, to get $x_t = (1-t)x_0 + \sqrt{t}\epsilon$, we can simply set $s(t) := 1-t$ and $\sigma(t) = \sqrt{t} / (1-t)$. In such a sense, I don’t think the claimed “decoupling the forward process into image-to-zero and zero-to-noise mappings” is a special feature of the proposed method.
- Section 4.2 can be largely improved by focusing on the continuous formulation of diffusion models. In specific, the reverse process can be directly derived from a forward SDE (Anderson, 1982), and this has become a more standard way as it makes everything more clear (see Song’s Score-SDE paper and Karras’ EDM paper). There is no need to use the “continuous-time Markov chain” and re-derive the conditional Gaussian distribution from the KL perspective.
- In the derivation of Eq (10) in Appendix A, I don’t know how you can obtain the second last equation in Eq (24). In particular, how do you evaluate the integral in the first term (from $t$ to $t - \Delta t$)? How do you get a grid of the cross-product term between ($h_{\theta} - h_t$ and $\epsilon_\theta - \epsilon$)?
- I am a little confused as to why the authors introduce $\phi_\theta$. It would be good to make it more precise how $\phi$ is related to $h_t$ and $x_0$.
- One claimed advantage of “simultaneously predicting the image and noise components” allows $x_t$ to be aligned with both starting and ending points at each time step. But there are already some papers (e.g. EDM https://arxiv.org/abs/2206.00364 and v-prediction https://arxiv.org/abs/2202.00512) that consider the better alignment between image and noise components in their training objective.
- In Section 3, the training objective Eq (4) is not exactly from (Ho et al., 2020). Precisely, Ho et al., 2020 didn’t consider the integral over t. Also, the denoising score matching objective should be $\nabla_x \log q(x_t | x_0)$ instead of $\nabla_x \log q(x_t )$.
- In experiments, I think many strong baselines are missing. For example, on CIFAR-10, what is the performance of EDM and DDIM? On CelebA-HQ-256, what is the performance of LDM? On Super-resolution and Image painting, what is the performance of other diffusion-based methods, such as Palette (https://arxiv.org/abs/2111.05826) and I2SB (https://arxiv.org/abs/2302.05872). In particular, I2SB claims to achieve the SOTA performance within 10 steps.
- Minor presentation issues: (Zhang & Chen, 2022) is not a distillation-based method (in the last paragraph in the introduction). In abstract and many other places, the authors imply that ODE samplers can result in less diverse samples. It is not quite clear to me. Any evidence or reference on this? In Figure 3, what do you mean by saying “sample generated images from $t=0.4$”? Also, which group of curves is about FID and which one is about MSE?

**Questions:**

See my comments in the above.